# Accelerating Molecular Graph Neural Networks via Knowledge Distillation

**Filip Ekström Kelvinius**[*]
Linköping University
`filip.ekstrom@liu.se`

**Dimitar Georgiev**[*]
Imperial College London
`d.georgiev21@imperial.ac.uk`

**Artur Petrov Toshev**[*]
Technical University of Munich
`artur.toshev@tum.de`

**Johannes Gasteiger**
Google Research
`johannesg@google.com`

## Abstract

Recent advances in graph neural networks (GNNs) have enabled more comprehensive modeling of molecules and molecular systems, thereby enhancing the precision of molecular property prediction and molecular simulations. Nonetheless, as the field has been progressing to bigger and more complex architectures, state-of-the-art GNNs have become largely prohibitive for many large-scale applications. In this paper, we explore the utility of knowledge distillation (KD) for accelerating molecular GNNs. To this end, we devise KD strategies that facilitate the distillation of hidden representations in directional and equivariant GNNs, and evaluate their performance on the regression task of energy and force prediction. We validate our protocols across different teacher-student configurations and datasets, and demonstrate that they can consistently boost the predictive accuracy of student models without any modifications to their architecture. Moreover, we conduct comprehensive optimization of various components of our framework, and investigate the potential of data augmentation to further enhance performance. All in all, we manage to close the gap in predictive accuracy between teacher and student models by as much as $96.7\%$ and $62.5\%$ for energy and force prediction respectively, while fully preserving the inference throughput of the more lightweight models.

## 1 Introduction

In the last couple of years, the field of molecular simulations has undergone a rapid paradigm shift with the advent of new, powerful computational tools based on machine learning (ML) [1]. At the forefront of this transformation have been recent advances in graph neural networks (GNNs), which have brought about architectures that more effectively capture geometric and structural information critical for the accurate representation of molecules and molecular systems [2, 3]. Consequently, a multitude of GNNs have been developed, which now offer predictive performance approaching that of conventional gold-standard methods such as density functional theory (DFT) at a fraction of the computational cost [4, 5, 6, 7, 8, 9]. This has, in turn, significantly accelerated the modeling of molecular properties and the simulation of molecular systems, bolstering new research developments in many scientific disciplines, including material sciences, drug discovery and catalysis [10, 11, 12, 13, 14].

Nonetheless, this progress - largely coinciding with the development of bigger and more complex models, has naturally come at the expense of increased complexity [15, 9, 7, 16]. This has gradually

---

[*]These authors contributed equally to this work. Order was determined by rolling a dice.

37th Conference on Neural Information Processing Systems (NeurIPS 2023).

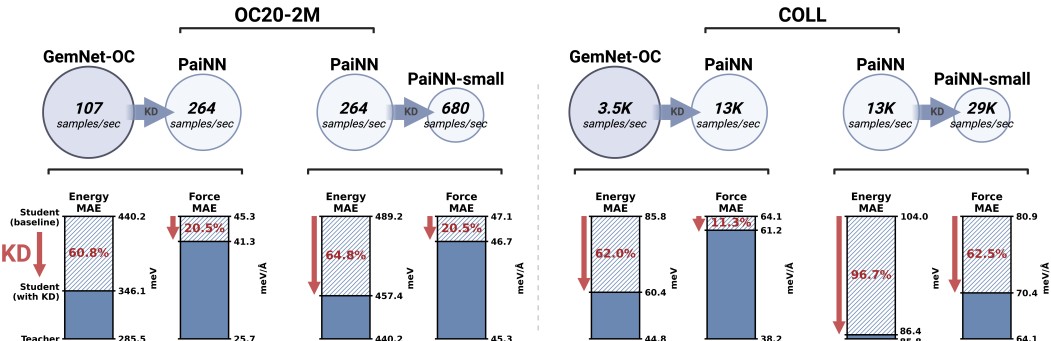

Figure 1: Using knowledge distillation, we manage to significantly boost the predictive accuracy of different student models on the OC20-2M [17] and COLL [6] datasets while fully preserving their inference throughput.

limited the utility of state-of-the-art GNNs in large-scale molecular simulation applications (e.g., molecular dynamics and high-throughput searches), where inference throughput (i.e., how many samples can be processed for a given time) is critical for making fast predictions about molecular systems at scale. Hence, addressing the trade-off between accuracy and computational demand remains essential for creating more affordable tools for molecular simulations and expanding the transformational impact of GNN models in the area.

Motivated by that, in this work, we investigate the potential of knowledge distillation (KD) in advancing the speed-accuracy Pareto frontier and enhancing the performance and scalability of molecular GNNs. In summary, the contributions of this paper are as follows:

- We, for the first time, explore the utility of knowledge distillation for accelerating GNNs for molecular simulations - a large-scale, multi-output regression task, challenging to address with common KD methods.
- We design custom KD strategies, which we call *node-to-node (n2n)*, *edge-to-edge (e2e)*, *edge-to-node (e2n)* and *vector-to-vector (v2v)* knowledge distillation, which facilitate the distillation of hidden representations in directional and equivariant molecular GNNs.
- We demonstrate the effectiveness of our protocols across different teacher-student configurations and datasets, allowing us to substantially improve the performance of student models while fully preserving their throughput (see Figure 1 for an overview).
- We conduct a comprehensive empirical analysis of different components of our KD strategies, as well as explore data augmentation techniques for further improving performance.

Associated code is available online[2].

## 2    Background

**Molecular simulations.** In this work, we consider molecular systems at an atomic level, i.e., $N$ atoms represented by their atomic numbers $\boldsymbol{z} = \{z_1, ..., z_N\} \in \mathbb{Z}^N$ and positions $\boldsymbol{X} = \{\boldsymbol{x}_1, \ldots, \boldsymbol{x}_N\} \in \mathbb{R}^{N \times 3}$. Given a system, we want a model that can predict the energy $E \in \mathbb{R}$ of the system, and the forces $\boldsymbol{F} \in \mathbb{R}^{N \times 3}$ acting on each atom. Both of these properties are of high interest when simulating molecular systems. The energy of a system is essential for the prediction of its stability, whereas the forces are important for molecular dynamics simulations, where computed forces are combined with the equations of motion to simulate the evolution of the system over time.

**GNNs for molecular systems.** GNNs are a suitable framework for modeling molecular systems. Each molecular system $(\boldsymbol{X}, \boldsymbol{z})$ can be represented as a mathematical graph $\mathcal{G} = (\mathcal{V}, \mathcal{E})$, where the nodes $\mathcal{V}$ correspond to the set of atoms, and edges $\mathcal{E}$ are created between nodes by connecting the closest neighboring atoms (typically defined by a cutoff radius and/or a maximum number of neighbors). Hence, in the context of molecular simulations, we can create GNNs that operate on

---

[2]https://github.com/gasteigerjo/ocp/blob/main/DISTILL.md

atomic graphs $\mathcal{G}$ by propagating information between the atoms and the edges, and make predictions about the energy and forces of each system in a multi-output manner - i.e., $\hat{E}, \hat{\boldsymbol{F}} = \text{GNN}(\boldsymbol{X}, \boldsymbol{z})$.

The main problem when modeling molecules and molecular properties is the number of underlying symmetries to account for, most importantly rigid transformations of the atoms. For instance, the total energy $E$ of a system is not affected by (i.e., is *invariant* to) rotations and translations of the system. However, the forces $\boldsymbol{F}$ do change as we rotate a system - i.e., they are *equivariant* to rotations. Therefore, to make accurate predictions about molecular systems, it is crucial to devise models that respect these symmetries and other physical constraints. There is now a plethora of diverse molecular GNNs that reflect that, e.g., SchNet [18], DimeNet [5, 6], PaiNN [19], GemNet [7, 8], NequIP [4], and SCN [9], which have incrementally established a more holistic description of molecular systems by capturing advanced geometric features and physical symmetries. This has, however, come at the expense of computational efficiency.

**Knowledge distillation.** Knowledge distillation is a technique for compressing and accelerating ML models [20], which has recently demonstrated significant potential in domains such as computer vision [21] and natural language modeling [22]. The main objective of KD is to create more efficient models by means of transferring knowledge (e.g., model parameters and activations) from large, computationally expensive, more accurate models, often referred to as teacher models, to simpler, more efficient models called student models [23]. Since the seminal work of Hinton *et al.* [24], the field has drastically expanded methodologically with the development of protocols that accommodate the distillation of "deeper" knowledge, more comprehensive transformation functions, as well as more robust distillation losses [23, 25]. Yet, these advances have mostly focused on classification, resulting in methods of limited utility in regression tasks [26]. Moreover, most research in the area has been confined to non-graph data (e.g., images, text, tabular data). Despite recent efforts to extend KD to graph data and GNNs, these have likewise only concentrated on classification tasks involving standard GNN architectures [27, 28]. And, in particular, the application of KD to large-scale regression problems in molecular simulations, which involve state-of-the-art molecular GNN architectures containing complex, geometric node- and edge-level features, is still unexplored.

## 3 Knowledge distillation for molecular GNNs

**Preliminaries.** In the context of the aforementioned prediction task, we train molecular GNNs by enforcing a loss $\mathcal{L}_0$ that combines both the energy and force prediction error as follows:

$$\mathcal{L}_0 = \alpha_{\text{E}} \mathcal{L}_{\text{E}}(\hat{E}, E) + \alpha_{\text{F}} \mathcal{L}_{\text{F}}(\hat{\boldsymbol{F}}, \boldsymbol{F}), \tag{1}$$

where $E$ and $\boldsymbol{F}$ are the ground-truth energy and forces, $\hat{E}$ and $\hat{\boldsymbol{F}}$ are the predictions of the model of interest, and $\mathcal{L}_{\text{E}}$ and $\mathcal{L}_{\text{F}}$ are some loss functions weighted by $\alpha_{\text{E}}, \alpha_{\text{F}} \in \mathbb{R}$.

To perform knowledge distillation, we augment this training process by defining an auxiliary KD loss term $\mathcal{L}_{\text{KD}}$, which is added to $\mathcal{L}_0$ (with a factor $\lambda \in \mathbb{R}$) to derive the final training loss function $\mathcal{L}$:

$$\mathcal{L} = \mathcal{L}_0 + \lambda \mathcal{L}_{\text{KD}}. \tag{2}$$

This was originally proposed in the context of classification by leveraging the fact that the soft label predictions (i.e., the logits after softmax normalization) of a given (teacher) model carry valuable information that can complement the ground-truth labels in the training process of another (student) model [24]. Since then, this has become the standard KD approach - commonly referred to as vanilla KD in the literature, which is often the foundation of new KD protocols. The main idea of this technique is to employ a KD loss $\mathcal{L}_{\text{KD}}$ that forces the student to mimic the predictions of the teacher model. This is usually achieved by constructing a loss $\mathcal{L}_{\text{KD}} = \text{KL}(z_s, z_t)$ based on the Kullback–Leibler (KL) divergence between the soft logits of the student $z_s$ and the teacher $z_t$.

However, such strategies - based on the distillation of the output of the teacher model only - pose two significant limitations. First, they are by design exclusively applicable to classification tasks, since there are no outputs analogous to logits in regression setups [20, 29]. This has consequently limited the utility of most KD methods in regression tasks. Second, this approach forces the student to emulate the final output of the teacher directly, which can be unattainable in regimes where the complexity gap between the two models is substantial, and thus detrimental to KD performance [30].

**Feature-based KD.** To circumvent these shortcomings, we focus on feature-based KD - an extension of vanilla KD concerned with the distillation of knowledge across the intermediate layers of models

[31, 32, 23]. In particular, we perform knowledge distillation of intermediate representations by devising a loss on selected hidden features $H_s \in U_s$ and $H_t \in U_t$ in the student and teacher models respectively, which takes the form:

$$\mathcal{L}_{\text{KD}} = \mathcal{L}_{\text{feat}}(\mathcal{M}_s(H_s), \mathcal{M}_t(H_t)), \tag{3}$$

where $\mathcal{M}_s : U_s \mapsto U$ and $\mathcal{M}_t : U_t \mapsto U$ are transformations that map the hidden features to a common feature space $U$, and $\mathcal{L}_{\text{feat}} : U \times U \mapsto \mathbb{R}^+$ is some loss of choice. The transformations $\mathcal{M}_s, \mathcal{M}_t$ can be the identity transformation, linear projections and multilayer perceptron (MLP) projection heads; whereas for the distillation loss $\mathcal{L}_{\text{feat}}$, typical options include mean squared error (MSE) and mean absolute error (MAE).

The fundamental design decision when devising a KD strategy based on the distillation of internal representations is the choice of features to distill between (i.e., features $H_s$ and $H_t$). One needs to ensure that paired features are similar both in expressivity and relevance to the output. Most research on feature-based distillation on graphs has so far focused on models that only have one type of scalar (node) features in classification tasks [27, 28], reducing the problem to the selection of layers to pair across the student and the teacher. This is often further simplified by utilizing models that share the same architecture up to reducing the number of blocks/layers and their dimensionality.

**Features in molecular GNNs.** In contrast, molecular GNNs contain diverse features (e.g., scalars, vectors and/or equivariant higher-order tensors based on spherical harmonics) organized across nodes and edges within a complex molecular graph. These are continually evolved by model-specific operators to infer molecular properties, such as energy and forces, in a multi-output prediction fashion. Therefore, features often represent different physical, geometric and/or topological information relevant to specific parts of the output. This significantly complicates the design of an effective KD strategy, especially when the teacher and the student differ architecturally, as one needs to extract and align representations corresponding to comparable features in both models.

In this work, we set out to devise KD strategies that are representative and effective across various molecular GNNs. This is why we investigate the effectiveness of KD with respect to GNNs that have distinct architectures and performance profiles, and can be organized in teacher-student configurations at different levels of architectural disparity. In particular, we employ the following three GNN models, ordered by computational complexity (ascending):

- *SchNet* [33]: A simple GNN model based on continuous-filter convolutional layers, which only contains scalar node features $s \in \mathbb{R}^d$. These are used to predict the energy $\hat{E}$. The force is then calculated as the negative gradient of the energy with respect to the atomic positions, i.e., $\hat{F} = -\nabla \hat{E}$.
- *PaiNN* [19]: A GNN based on equivariant message passing, which contains scalar node features $x \in \mathbb{R}^{d_1}$ - used for energy prediction; as well as geometric vectorial node features $v \in \mathbb{R}^{3 \times d_2}$ that are equivariant to rotations and can thus be combined with the scalar features to make direct predictions of the forces (i.e., without computing gradients of the energy).
- *GemNet-OC* [8]: A GNN model that utilizes directional message passing between scalar node features $h \in \mathbb{R}^{d_h}$ and scalar edges features $m \in \mathbb{R}^{d_m}$. After each block of layers, these are processed through an output block, resulting in scalar node features $x_E^{(i)}$ and edge features $x_F^{(i)}$, where $i$ is the block number. The output features from each block are aggregated into output features $x_E$ and $x_F$, which are used to compute the energy and forces respectively.

An overview of the features of the three models can be found in Table 1.

Table 1: An overview of the types of features available in the three models we use in this study.

|  | SchNet | PaiNN | GemNet-OC |
|---|:---:|:---:|:---:|
| Scalar node features | ✓ | ✓ | ✓ |
| Scalar edge features |  |  | ✓ |
| Vectorial node features |  | ✓ |  |
| Output blocks |  |  | ✓ |

**Defining feature-based KD distillation strategies for molecular GNNs.** In the context of the three models considered in this work, we devise the following KD strategies:

- *node-to-node (n2n):* As all three models contain scalar node features $H_{\text{node}}$, we can distill knowledge in between these directly by defining a loss $\mathcal{L}_{KD}$ given by

$$\mathcal{L}_{KD} = \mathcal{L}_{\text{feat}}(\mathcal{M}_s(H_{\text{node,s}}), \mathcal{M}_t(H_{\text{node,t}})), \tag{4}$$

where $H_{\text{node,s}}$ and $H_{\text{node,t}}$ represent the node features of the student and teacher, respectively. Note this is a general approach that utilizes scalar node features only, making it applicable to standard GNNs. Here, we want to force the student to mimic the representations of the teacher for each node (i.e., atom) independently, so we use a loss that directly penalizes the distance between the features in the two models, such as MSE (similar to the original formulation of feature-based KD in Romero *et al.* [31]). Other recently proposed losses $\mathcal{L}_{\text{feat}}$ for the distillation of node features in standard GNNs specifically include approaches based on contrastive learning [34, 35, 36, 37] and adversarial training [38]. We do not focus on such methods as much since they are better suited for (node) classification tasks (e.g., contrasting different classes of nodes), and not for molecule-level predictions.

To take advantage of other types of features relevant to molecular GNNs, we further devise three additional protocols, which we outline below.

- *edge-to-edge (e2e):* The GemNet-OC model heavily relies on its edge features, which are a key component of the directional message passing employed in the architecture. As such, they can be a useful resource for KD. Hence, we also consider KD between edge features, which we accomplish by applying Equation (4) to the edge features $H_{\text{edge,s}}$ and $H_{\text{edge,t}}$ of the student and teacher, respectively.

- *edge-to-node (e2n):* However, not all models considered in this study contain edge features to distill to. To accommodate that, we propose a KD strategy where we transfer information from GemNet-OC's edge features $H_{\text{edge},(i,j)}$ by first aggregating them as follows:

$$H_{\text{edge2node},i} = \sum_{j \in \mathcal{N}(i)} H_{\text{edge},(i,j)}, \tag{5}$$

where $i$ is some node index. The resulting features $H_{\text{edge2node},i}$ are scalar, node-level features, and we can, therefore, use them to transfer knowledge to the student node features $H_{\text{node,s}}$ as in Equation (4).

- *vector-to-vector (v2v):* Similarly, the PaiNN model defines custom vectorial node features, which differ from the scalar (node and edge) features available in the other models. These are not scalar and invariant to rigid transformations of the atoms, but geometrical vectors that are equivariant with respect to rotations. As these carry important information about a given system, we also want to define a procedure to distill these. When we perform KD between two PaiNN models, we can directly distill information between these vectorial features just as in Equation (4). In contrast, when distilling knowledge into PaiNN from our GemNet-OC teacher that has no such vectorial features, we transfer knowledge between (invariant) scalar edge features and (equivariant) vectorial node features by noting that scalar edge features sit on an equivariant 3D grid since they are associated with an edge between two atoms in 3D space. Hence, we can aggregate the edge features $\{H_{\text{edge},(i,j)}\}_{j \in \mathcal{N}}$ corresponding to a given node $i$ into node-level equivariant vectorial features $H_{\text{vec},i}$ by considering the unit vector $\boldsymbol{u}_{ij} = \frac{1}{|\boldsymbol{x}_j - \boldsymbol{x}_i|}(\boldsymbol{x}_j - \boldsymbol{x}_i)$ that defines the direction of the edge $(i,j)$, such that

$$H_{\text{vec},i}^{(k)} = \sum_{j \in \mathcal{N}(i)} \boldsymbol{u}_{ij} H_{\text{edge},(i,j)}^{(k)}, \tag{6}$$

with the superscript $k$ indicating the channel. Notice that the features $H_{\text{vec},i}^{(k)}$ fulfill the condition of equivariance with respect to rotations as each vector $\boldsymbol{u}_{ij}$ is equivariant to rotations, and $H_{\text{edge},(i,j)}^{(k)}$ - a scalar not influencing its direction. Consequently, it is important to use a loss $\mathcal{L}_{\text{feat}}$ that encourages vectors to align in both magnitude and direction - e.g., MSE.

**Additional KD strategies.** We further evaluate two additional KD approaches inspired by the vanilla logit-based KD used in classification, which we augment to make suitable for regression tasks:

- *Vanilla (1):* One way of adapting vanilla KD for regression is by steering the student to mimic the final output of the teacher directly:

$$\mathcal{L}_{\text{KD}} = \alpha_{\text{E}} \mathcal{L}_{\text{E}}(\hat{E}_s, \hat{E}_t) + \alpha_{\text{F}} \mathcal{L}_{\text{F}}(\hat{\boldsymbol{F}}_s, \hat{\boldsymbol{F}}_t), \tag{7}$$

where the subscripts $_s$ and $_t$ refer to the predictions of the student and teacher, respectively. Note that, unlike in classification, this approach does not provide much additional information in regression tasks, except for some limited signal about the error distribution of the teacher model [20, 29].

- *Vanilla (2):* One way to enhance the teacher signal during training is to consider the fact that many GNNs for molecular simulations make separate atom- and edge-level predictions, which are consequently aggregated into a final output. For instance, the total energy $E$ of a system is usually defined as the sum of the predicted contributions from each atom $\hat{E} = \sum_i \hat{E}_i$. Hence, we can extend the aforementioned vanilla KD approach by imposing a loss on these granular predictions instead:

$$\mathcal{L}_{\text{KD}} = \frac{1}{N} \sum_{i=1}^{N} \mathcal{L}_{\text{E}}(\hat{E}_{i,s}, \hat{E}_{i,t}).\tag{8}$$

These individual energy contributions are not part of the labeled data, but, when injected during training, can provide more fine-grained information than the aggregated prediction.

## 4 Experimental results

To evaluate our proposed methods, we perform comprehensive benchmarking experiments on the OC20-2M [17] dataset (structure to energy and forces (S2EF) task) - a large and diverse catalyst dataset; and COLL [6] - a challenging molecular dynamics dataset. We use the model implementations provided in the Open Catalyst Project (OCP) codebase[3] (see Appendix A for detailed information about training procedure and model hyperparameters).

**Benchmarking baseline models.** We start by first evaluating the baseline performance of the models we employ in this study. As previously mentioned, we select SchNet, PaiNN and GemNet-OC for our experiments as they cover most of the accuracy-complexity spectrum, with the last representing the state-of-the-art on OC20 S2EF and COLL at the time of experimentation. To demonstrate this, we benchmark the predictive accuracy and inference throughput of the models on the two aforementioned datasets. In conjunction with the default PaiNN and GemNet-OC models, we also experiment with more lightweight versions of the two architectures - referred to as PaiNN-small and GemNet-OC-small respectively, where we reduce the number of hidden layers and their dimensionality. We train all models to convergence ourselves, except for the GemNet-OC model on OC20-2M, where we utilize the pre-trained model available within the OCP repository (July 2022).

We present our benchmarking results on OC20 S2EF in Table 2, which summarizes the performance of the five models with respect to the following four metrics: energy and force MAE (i.e., the mean absolute error between ground truth and predicted energies and forces); force cos (i.e., the cosine similarity between ground truth and predicted forces); and energy and forces within threshold (EFwT) - i.e., the percentage of systems whose predicted energies and forces are within a specified threshold from the ground truth [17]. Since force cos and EFwT are correlated with energy and force MAE, we focus on the latter throughout the paper but present all four for completeness.

Table 2: Evaluation of the performance of our five baseline models on the OC20 S2EF task. All models are trained on the OC20-2M dataset. Values represent the average across the four available validation sets. Results for individual validation datasets are provided in Appendix B.

| Model | Inference Throughput Samples / GPU sec. ↑ | OC20 S2EF Validation Energy MAE meV ↓ | Force MAE meV/Å ↓ | Force cos ↑ | EFwT % ↑ |
|---|---|---|---|---|---|
| SchNet | 1100 | 1308 | 65.1 | 0.204 | 0 |
| PaiNN-small | 680 | 489 | 47.1 | 0.345 | 0.085 |
| PaiNN | 264 | 440 | 45.3 | 0.376 | 0.14 |
| GemNet-OC-small | 158 | 344 | 31.3 | 0.524 | 0.51 |
| GemNet-OC | 107 | 286 | 25.7 | 0.598 | 1.06 |

[3]https://github.com/Open-Catalyst-Project/ocp

Our results highlight the substantial trade-off between predictive accuracy and computational cost across the GNN architectures, and, therefore, the need for methods that can alleviate this limitation. We observe the same trend on the COLL dataset (see Appendix C).

**Similarity analysis of baseline models.** To make our analysis exhaustive, we set out to design experiments involving teacher and student architectures of a variable degree of architectural disparity. As a proxy of that, we derive similarity scores based on central kernel alignment (CKA) [39, 40, 35]. In particular, we calculate the pairwise CKA similarity between the node features of our trained SchNet, PaiNN and GemNet-OC models. The results of this analysis are summarized in Figure 2. Focusing on intra-model similarities first (plots on the diagonal), we observe that, while representations from different layers within PaiNN and SchNet have a generally high degree of similarity, GemNet-OC exhibits the opposite behavior, with features extracted at each layer being significantly different from those captured across the rest of the architecture. This is consistent with the architectures of these three models, with features in PaiNN and SchNet being iteratively updated by adding

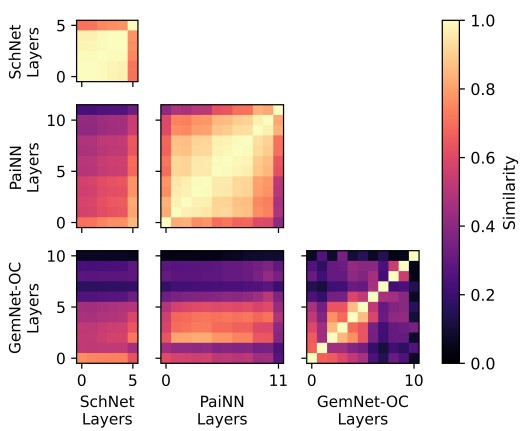

Figure 2: Similarity analysis between the node features of SchNet, PaiNN and GemNet-OC using CKA (averaged over $n = 987$ nodes).

displacement features computed at each layer, while those in GemNet-OC representing separate output features. When examining inter-model similarity instead, we notice that, generally speaking, node features in SchNet and PaiNN are similar, whereas those between SchNet and GemNet-OC, and PaiNN and GemNet-OC, diverge significantly as we move deeper into GemNet-OC.

**Knowledge distillation results.** Based on the aforementioned analyses, we define the following teacher-student pairs, covering the whole spectrum of architectural disparity: PaiNN to PaiNN-small (*same* architecture); PaiNN to SchNet (*similar* architectures); GemNet-OC to PaiNN (*different* architectures). We additionally explore KD from GemNet-OC to GemNet-OC-small (*same* architecture) on OC20. We train student models by utilizing an offline KD strategy [23], where we distill knowledge from the more competent, pre-trained teacher model to the simpler, more lightweight student model during the training of the latter. We augment the training of each student model with our KD protocols and evaluate the effect on predictive accuracy against the models trained without KD. If not mentioned otherwise, we utilize the following setup: we use MSE as a distillation loss $\mathcal{L}_{\text{feat}}$; a learned linear layer as a transformation function $\mathcal{M}_s$ applied to the features of the student; and the identity transformation as $\mathcal{M}_t$. When distilling knowledge from/into GemNet-OC models, we use the aggregated node- and edge-level output features, which is reminiscent of the review-based setup proposed in [41]. For PaiNN and SchNet, we use the final node features.

The results of our experiments are summarized in Tables 3 and 4, presenting a comparative analysis of the predictive performance of different student models trained with and without the implementation of knowledge distillation on the OC20-2M and COLL datasets, respectively. Focusing on energy predictions first, we observe that, by utilizing KD, we achieve significant improvements in performance in virtually all teacher-student configurations. In particular, we manage to close the gap in performance between student and teacher models by $\sim 60\%$ or more in six out of the seven configurations, reaching results as high as $96.7\%$ (distilling PaiNN to PaiNN-small on the COLL dataset). Putting our results into context, we remark that our PaiNN model trained with *n2n* KD from GemNet-OC, for instance, provides more accurate energy predictions than more advanced models such as GemNet-dT [7, 8] which is substantially slower. The only setup where results are not as definite is when training SchNet on OC20-2M with KD from PaiNN, where we close $10.8\%$. However, it is noteworthy to highlight that this still corresponds to a significant absolute improvement (i.e., notice the big initial difference in performance between the two baseline models on this dataset).

We similarly observe an improvement in the accuracy of student models in force predictions in all teacher-student configurations. Although we observe force improvements (typically $\sim$5–25%) that are generally not as pronounced as those achieved in energy prediction, we note that we reach

Table 3: Evaluation of the performance of our KD strategies across teacher-student architectures on the OC20 S2EF task. All models are trained on the OC20-2M dataset. Numbers in brackets represent the proportion of the gap between the student ($S$) and the teacher ($T$) that has been closed by the respective KD strategy (in %). Best results are given in **bold**. Values represent the average across the four available validation sets. Results for individual validation datasets are provided in Appendix B. Error bars for selected configurations can be found in Appendix F.

| | Model | OC20 S2EF Validation | | | |
| | | Energy MAE meV ↓ | Force MAE meV/Å ↓ | Force cos ↑ | EFwT % ↑ |
|---|---|---|---|---|---|
| *same* | *S*: PaiNN-small | 489 | 47.1 | 0.345 | 0.085 |
| | *T*: PaiNN | 440 | 45.3 | 0.376 | 0.139 |
| | *Vanilla (1)* | 515(-52.4%) | 48.5(-81.0%) | 0.269(-237%) | 0.07(-28%) |
| | *Vanilla (2)* | 476(27.2%) | 50.8(-215%) | 0.307(-117%) | 0.068(-32.6%) |
| | *n2n* | **457 (64.8%)** | **46.7 (20.5%)** | **0.348 (9.3%)** | **0.085 (0.5%)** |
| | *v2v* | 459(60.8%) | 47.2(-9.1%) | 0.347(6.8%) | 0.079(-11.9%) |
| | *S*: GemNet-OC-small | 344 | 31.3 | 0.524 | 0.51 |
| | *T*: GemNet-OC | 286 | 25.7 | 0.598 | 1.063 |
| | *Vanilla (1)* | 339(9.1%) | 31.2(-1.0%) | 0.525(1.3%) | 0.51(0.4%) |
| | *Vanilla (2)* | 328(27.8%) | 31.2(0.3%) | 0.525(1.5%) | **0.61 (18.3%)** |
| | *n2n* | **310 (58.8%)** | 31.1(2.2%) | 0.526(3.6%) | 0.61(18.2%) |
| | *e2e* | 334(16.5%) | **29.7 (27.6%)** | **0.543 (25.7%)** | 0.58(12.6%) |
| *similar* | *S*: SchNet | 1308 | 65.1 | 0.204 | 0 |
| | *T*: PaiNN | 440 | 45.3 | 0.376 | 0.139 |
| | *Vanilla (1)* | **1214 (10.8%)** | 64.6(2.3%) | **0.230 (15.2%)** | **0.003 (1.8%)** |
| | *Vanilla (2)* | 1216(10.5%) | **64.6 (2.5%)** | 0.229(14.5%) | 0(0%) |
| | *n2n* | 1251(6.6%) | 65.2(-0.5%) | 0.223(11.1%) | 0(0%) |
| *different* | *S*: PaiNN | 440 | 45.3 | 0.376 | 0.139 |
| | *T*: GemNet-OC | 286 | 25.7 | 0.598 | 1.063 |
| | *Vanilla (1)* | 440(0.0%) | 43.9(7.1%) | 0.378(0.8%) | 0.14(0.4%) |
| | *Vanilla (2)* | 419(13.6%) | 114.8(-353%) | 0.324(-23.8%) | 0.127(-1.3%) |
| | *n2n* | **346 (60.8%)** | 42.8(12.8%) | 0.393(7.4%) | **0.262 (13.4%)** |
| | *e2n* | 418(14.2%) | **41.3 (20.5%)** | **0.405 (12.8%)** | 0.207(7.4%) |
| | *v2v* | 437(1.8%) | 42.9(17.1%) | 0.397(9.4%) | 0.124(-1.6%) |

results as high as 62.5% for some configurations (distilling PaiNN to PaiNN-small on COLL). A possible reason for this difference in improvement between energy and forces could be attributed to the nature of the supervised task - there are substantially more force labels (i.e., one 3D vector per atom, which could be hundreds per sample) than energy labels (i.e., one per sample). Consequently, we hypothesize it is easier for models to learn to make accurate force predictions, and, therefore, there is more room for improvement in the energy predictions, which we can target with KD.

All in all, our experiments demonstrate that, by applying knowledge distillation, we successfully enhance the performance of student models across all teacher-student configurations and datasets, confirming the effectiveness and robustness of the approach in the context of molecular GNNs. We reiterate the fact that no modifications to the student architectures are made, meaning we achieve an out-of-the-box boost in accuracy without impacting inference throughput.

**Effect of KD on model similarity.** We continue our investigation by exploring how KD affects student models and their features. To this end, we analyze how the CKA similarity between teacher and student models changes with the introduction of KD during training. We present the outcome of one of our analyses in Figure 3, where we summarize how the similarity between the node features of GemNet-OC and PaiNN changes with the implementation of *n2n* KD. We observe that KD introduces strong and specific similarity gains in the layers we use for KD, which also propagates along the

Table 4: Evaluation results on the COLL test set. Numbers in brackets represent the proportion of the gap between the student ($S$) and the teacher ($T$) that has been closed by the respective KD strategy (in %). Best results are given in **bold**.

| | Model | COLL test set | | | |
| | | Energy MAE meV ↓ | Force MAE meV/Å ↓ | Force cos ↑ | EFwT % ↑ |
|---|---|---|---|---|---|
| *same* | *S*: PaiNN-small | 104.0 | 80.9 | 0.984 | 5.4 |
| | *T*: PaiNN | 85.8 | 64.1 | 0.988 | 10.1 |
| | *Vanilla (1)* | 106.1(-11.5%) | 82.0(-6.5%) | 0.984(2.3%) | 4.46(-20.2%) |
| | *Vanilla (2)* | **86.4 (96.7%)** | 80.9(0%) | 0.983(-2.3%) | 4.3(-23.7%) |
| | *n2n* | 92.5(63.2%) | 77.8(18.5%) | 0.984(18.2%) | **6.63 (26.5%)** |
| | *v2v* | 90.4(74.7%) | **70.4 (62.5%)** | **0.986 (45.5%)** | 5.8(8.4%) |
| *similar* | *S*: SchNet | 146.5 | 121.2 | 0.970 | 2.75 |
| | *T*: PaiNN | 85.8 | 64.1 | 0.988 | 10.1 |
| | *Vanilla (1)* | 146.1(0.7%) | 120.8(0.7%) | 0.970(1.1%) | 2.54(-2.9%) |
| | *Vanilla (2)* | **104.1 (69.9%)** | 120.9(0.5%) | 0.970(1.1%) | **6.45 (50.7%)** |
| | *n2n* | 141.6(8.1%) | **117.2 (7.0%)** | **0.971 (5.4%)** | 2.63(-1.6%) |
| *different* | *S*: PaiNN | 85.8 | 64.1 | 0.988 | 10.1 |
| | *T*: GemNet-OC | 44.8 | 38.2 | 0.994 | 20.2 |
| | *Vanilla (1)* | 86.2(-1.1%) | 63.9(0.6%) | 0.988(1.5%) | 10.1(0.1%) |
| | *Vanilla (2)* | 61.4(59.5%) | 62.9(4.6%) | 0.988(5.2%) | 13.0(29.2%) |
| | *n2n* | **60.4 (62.0%)** | **61.2 (11.3%)** | **0.989 (14.9%)** | **13.6 (34.6%)** |
| | *e2n* | 77.3(20.8%) | 63.3(3.0%) | 0.988(7.9%) | 11.0(9.2%) |
| | *v2v* | 81.2(11.2%) | 63.3(3.1%) | 0.988(3.4%) | 10.5(4.6%) |

student architecture. We notice similar behavior across other teacher-student configurations and KD strategies (see Figures 6 and 7 in Appendix G), allowing us to monitor and quantify the effect of KD on student models as we explore different KD settings and design choices.

**Hyperparameter studies.** We additionally conduct a thorough hyperparameter study to evaluate the effect of different design choices within our KD framework. We summarize our results below.

*- Effect of distillation loss*: Apart from our default MSE-based distillation loss $\mathcal{L}_{\text{feat}}$, we also experimented with more advanced losses such as Local Structure Preservation (LSP) [34] and Global Structure Preservation (GSP) [35], as well as directly optimizing CKA. We observed the best results with our default MSE loss, with other options substantially hurting accuracy (see Appendix E.1).

*- Effect of transformation function*: We also investigated a number of different transformation functions and the effect they have on performance. The transformations we utilized include: the identity transformation (when appropriate); learned linear transformations, and MLP projection heads. Our results showed that our default linear mapping is a sufficiently flexible choice as it gives the best results(see Appendix E.2).

*- Effect of feature selection*: We additionally explored the effect of feature selection on KD performance. In particular, we analyzed the change in the predictive accuracy of PaiNN as a student model as we distill features from earlier layers in the teacher (GemNet-OC), or distill knowledge into earlier layers in the student. Our results suggest that using features closer to the output is the best-performing strategy (see Appendix E.3). We also performed CKA-based similarity analyses to monitor how model similarity changes as we vary the features we used for KD (see Figure 8 in Appendix G).

**Data augmentation.** As for most other applications, data labeling for molecular data is costly as it requires running computationally expensive quantum mechanical calculations to obtain ground truth energies and forces. Motivated by this, we explore two data augmentation techniques, which we use to generate new data points that we label with the teacher and use for KD. We briefly describe these below (see Appendix D for more details).

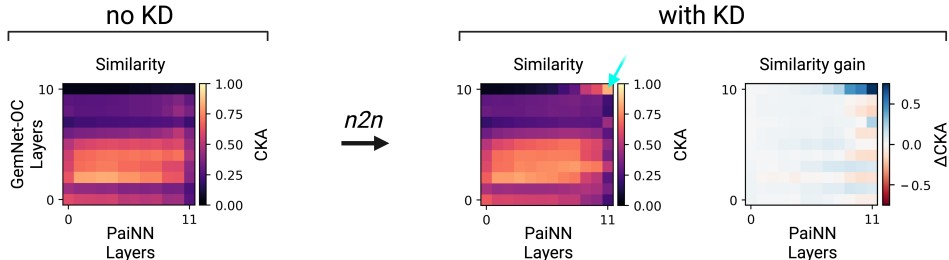

Figure 3: Similarity analysis between Gemnet-OC and PaiNN without KD (left) and with KD (right). The feature pair used during KD is indicated with ⟋. Similarity analyses for other KD strategies and teacher-student configurations are presented in Appendix G.

*- Random rattling:* Adding noise to existing structures (also known as "rattling") is a form of data augmentation that has been used in the context of pretraining of molecular GNNs [42, 43], and as a regularization strategy [44]. Inspired by this, we utilized "rattling" in the context of KD, where we added random noise to the atomic positions of systems and used the teacher to derive energy and force labels for these rattled samples. We then combined these rattled structures with the original dataset during the training of student models. However, this approach did not provide significant improvements. Additionally, we tried using gradient ascent to find perturbations that maximize the discrepancy between the teacher and student predictions, similar to [45], but this did not show improvements over random noise, and also increased training time.

*- Synthetic Data:* Samples in OC20 S2EF originate from the same relaxation trajectory and are therefore correlated. To tackle this, we generated our own distilled dataset coined *d1M*, which consists of one million samples generated by sampling new systems (generated with the OC Datasets codebase [4]), running relaxations with our pre-trained GemNet-OC model, and then subsampling approximately $10\%$ of the frames. We explored different ways of incorporating this new *d1M* dataset, all of which were based on joint training with the OC20 S2EF 2M data (similar to what we did with the rattled systems). To study different combinations of the ground truth DFT samples and the *d1M* samples during training, we defined two hyperparameters determining: (a) how many of the samples per batch originate from each of the datasets; and (b) how to weight the loss contributions based on the origin of data. Unfortunately, and contrary to similar approaches, e.g., in speech recognition [46], the results we observed did not significantly improve on the baseline models.

## 5    Conclusion

In this paper, we investigated the utility of knowledge distillation in the context of GNNs for molecules. To this end, we proposed four distinct feature-based KD strategies, which we validated across different teacher-student configurations and datasets. We showed that our KD protocols can significantly enhance the performance of different molecular GNNs without any modifications to their architecture, allowing us to run faster molecular simulations without substantially impairing predictive accuracy. With this work, we aim to elucidate the potential of KD in the domain of molecular GNNs and stimulate future research in the area. Interesting future directions include: the combination of KD strategies (e.g., *n2n* and *v2v*); extending the framework to other types of features (e.g., tensorial features [47]), molecular tasks and datasets; better understanding the connection between KD performance and model expressivity (e.g., can model similarity inform KD design); and performing a more comprehensive stability analysis [48]. One caveat of our approach is that even though inference times are not affected, training times are, albeit not necessarily if a pre-trained teacher model is available (see Appendix H). Finally, it is important to recognize that such technologies, while innovative, could be used for potentially harmful purposes, such as the simulation or discovery of toxic systems, or the development of harmful technologies.

---

[4]https://github.com/Open-Catalyst-Project/Open-Catalyst-Dataset

## Acknowledgments and Disclosure of Funding

F.E.K. is financially supported by the Excellence Center at Linköping–Lund in Information Technology (ELLIIT). D.G. is supported by UK Research and Innovation [UKRI Centre for Doctoral Training in AI for Healthcare grant number EP/S023283/1]. Computing resources provided by: the Berzelius resource at the National Supercomputer Centre, provided by Knut and Alice Wallenberg Foundation; the Alvis resource provided by the National Academic Infrastructure for Supercomputing in Sweden (NAISS) at Chalmers e-Commons at Chalmers (C3SE) partially funded by the Swedish Research Council through grant agreement no. 2022-06725; the Chair of Aerodynamics and Fluid Mechanics at Technical University of Munich. This research project was initially conceived at the 2022 LOGML summer school, and we would like to thank Guocheng Qian and I-Ju Chen for their contribution during the early conceptualizing stages of this project during and in the first weeks following the summer school. We also thank the Open Catalyst team for their open-source codebase, support and discussions. In particular, Muhammed Shuaibi for providing the COLL dataset in LMDB format. Figures assembled in BioRender.

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

# A   Training and hyperparameters

Models were trained on NVIDIA A100 40 GB and NVIDIA RTX A6000 48 GB GPUs, except GemNet-OC-small which were trained on NVIDIA A100 80 GB and NVIDIA RTX A6000 48 GB. All models were trained on single GPUs, except for SchNet when trained on OC20-2M, which required 3 GPUs. Inference throughput was profiled on A100 40 GB GPUs, with reported values representing approximate numbers averaged across three evaluations. We provide detailed information about the hyperparameters we used for each model in Tables 5, 6, and 7.

Moreover, we summarize the KD weighting factors $\lambda$ we used for each model configuration in Table 8.

Table 5: SchNet hyperparameters.

| Hyperparameter | OC20 | COLL |
|---|---|---|
| Hidden channels | 1024 | 128 |
| Filters | 256 | 128 |
| Interaction blocks | 5 | 6 |
| Gaussians | 200 | 50 |
| Cutoff | 6.0 | 12.0 |
| | | |
| Batch size | 192 | 32 |
| Initial learning rate | $10^{-4}$ | $10^{-3}$ |
| Optimizer | AdamW | AdamW |
| Scheduler | LambdaLR | LinearWarmupExponentialDecay |
| Learning rate decay factor | 0.1 | 0.01 |
| Learning rate milestones | 52083, 83333, 104166 | - |
| Warmup steps | 31250 | 3750 |
| Warmup factor | 0.1 | - |
| Force Coefficient | 100 | 100 |
| Energy Coefficient | 1 | 1 |
| Number of epochs | 30 | 500 |

Table 6: PaiNN hyperparameters. Slash-separated values indicate PaiNN versus PaiNN-small hyperparameters.

| Hyperparameter | OC20 | COLL |
|---|---|---|
| Hidden channels | 512/256 | 256/128 |
| Number of layers | 6/4 | 6/4 |
| Number of RBFs | 128 | 128 |
| Cutoff | 12.0 | 12.0 |
| Max. num. neighbors | 50 | 50 |
| Direct Forces | True | True |
| | | |
| Batch size | 32 | 32 |
| Optimizer | AdamW | AdamW |
| AMSGrad | True | True |
| Initial learning rate | $10^{-4}$ | $10^{-3}$ |
| Scheduler | LambdaLR | LinearWarmupExponentialDecay |
| Warmup steps | None | 3750 |
| Learning rate decay factor | 0.45 | 0.01 |
| Learning rate milestones (steps) | 160000, 320000, 480000, 640000 | - |
| Force coefficient | 100 | 100 |
| Energy coefficient | 1 | 1 |
| EMA decay | 0.999 | 0.999 |
| Gradient clip norm threshold | 10 | 10 |
| Epochs | 16 | 375 |

Table 7: GemNet-OC hyperparameters. Slash-separated values indicate GemNet-OC versus GemNet-OC-small hyperparameters.

| Hyperparameter | OC20 | COLL |
|---|---|---|
| No. spherical basis | 7 | 7 |
| No. radial basis | 128 | 128 |
| No. blocks | 4/3 | 4 |
| Atom embedding size | 256/128 | 128 |
| Edge embedding size | 512/256 | 256 |
| | | |
| Triplet edge embedding input size | 64 | 64 |
| Triplet edge embedding output size | 64 | 64 |
| Quadruplet edge embedding input size | 32 | 32 |
| Quadruplet edge embedding output size | 32 | 32 |
| Atom interaction embedding input size | 64 | 64 |
| Atom interaction embedding output size | 64 | 64 |
| Radial basis embedding size | 16 | 16 |
| Circular basis embedding size | 16 | 16 |
| Spherical basis embedding size | 32 | 32 |
| | | |
| No. residual blocks before skip connection | 2 | 2 |
| No. residual blocks after skip connection | 2 | 2 |
| No. residual blocks after concatenation | 1 | 1 |
| No. residual blocks in atom embedding blocks | 3 | 3 |
| No. atom embedding output layers | 3 | 3 |
| | | |
| Cutoff | 12.0 | 12.0 |
| Quadruplet cutoff | 12.0 | 12.0 |
| Atom edge interaction cutoff | 12.0 | 12.0 |
| Atom interaction cutoff | 12.0 | 12.0 |
| Max interaction neighbors | 30 | 30 |
| Max quadruplet interaction neighbors | 8 | 8 |
| Max atom edge interaction neighbors | 20 | 20 |
| Max atom interaction neighbors | 1000 | 1000 |
| | | |
| Radial basis function | Gaussian | Gaussian |
| Circular basis function | Spherical harmonics | Spherical Harmonics |
| Spherical basis function | Legendre Outer | Legendre Outer |
| Quadruplet interaction | True | True |
| Atom edge interaction | True | True |
| Edge atom interaction | True | True |
| Atom interaction | True | True |
| Direct forces | True | True |
| | | |
| Activation | Silu | Silu |
| Optimizer | AdamW | AdamW |
| Scheduler | ReduceLROnPlateau | LinearWarmup ExponentialDecay |
| Force coefficient | 100 | 100 |
| Energy coefficient | 1 | 1 |
| EMA decay | 0.999 | 0.999 |
| Gradient clip norm threshold | 10 | 10 |
| Initial learning rate | $5 \times 10^{-4}$ | $10^{-3}$ |
| Epochs | 80/9 | 165 |

Table 8: Choice of the weighting factor $\lambda$ of the KD loss for the different teacher-student configurations and KD strategies.

| Teacher | Student | KD | OC20 | COLL |
|---------|---------|-----|------|------|
| GemNet-OC | PaiNN | *Vanilla (1)* | 1.0 | 0.2 |
| GemNet-OC | PaiNN | *Vanilla (2)* | 500 | 100 |
| GemNet-OC | PaiNN | *n2n* | 10000 | 1000 |
| GemNet-OC | PaiNN | *e2n* | 1000 | 10 |
| GemNet-OC | PaiNN | *v2v* | 50000 | 100 |
| | | | | |
| GemNet-OC | GemNet-OC-small | *Vanilla (1)* | 0.2 | - |
| GemNet-OC | GemNet-OC-small | *Vanilla (2)* | 10.0 | - |
| GemNet-OC | GemNet-OC-small | *n2n* | 1000.0 | - |
| GemNet-OC | GemNet-OC-small | *e2e* | 100000 | - |
| | | | | |
| PaiNN | PaiNN-small | *Vanilla (1)* | 1 | 1 |
| PaiNN | PaiNN-small | *Vanilla (2)* | 200 | 100 |
| PaiNN | PaiNN-small | *n2n* | 100 | 100 |
| PaiNN | PaiNN-small | *v2v* | 1000 | 10000 |
| | | | | |
| PaiNN | SchNet | *Vanilla (1)* | 0.1 | 1 |
| PaiNN | SchNet | *Vanilla (2)* | 0.1 | 100 |
| PaiNN | SchNet | *n2n* | 1000 | 100 |

# B    Full validation results on OC20

Tables 9-12 present the extended results on OC20 across the 4 separate S2EF validation sets.

Table 9: Evaluation results on the OC20 S2EF **in-distribution** validation set.

| | Model | Energy MAE meV ↓ | Force MAE meV/Å ↓ | Force cos ↑ | EFwT % ↑ |
|---|---|---|---|---|---|
| | *S*: PaiNN-small | 409 | 41.6 | 0.357 | 0.16 |
| | *T*: PaiNN | 358 | 38.5 | 0.390 | 0.25 |
| *same* | *Vanilla (1)* | 426(-33.7%) | 42.6(-32.3%) | 0.35(-21.9%) | 0.12(-44.4%) |
| | *Vanilla (2)* | 396(25.7%) | 45.7(-132.3%) | 0.316(-126.9%) | 0.11(-55.6%) |
| | *n2n* | **393 (31.2%)** | 41.7(-2.5%) | **0.359 (4.7%)** | 0.15(-7.8%) |
| | *v2v* | 406(5.6%) | 42.1(-16.9%) | 0.357(-2.6%) | 0.13(-32.6%) |
| | *S*: GemNet-OC-small | 292 | 27.7 | 0.534 | 0.90 |
| | *T*: GemNet-OC | 226 | 22.5 | 0.610 | 1.09 |
| | *Vanilla (1)* | 292(0.1%) | 27.7(0.0%) | 0.535(1.1%) | 0.92(1.8%) |
| | *Vanilla (2)* | 283(13.5%) | 27.7(-0.4%) | 0.535(0.1%) | 1.01(18.8%) |
| | *n2n* | **252 (61.1%)** | 27.5(3.8%) | 0.536(1.6%) | **1.09 (19.2%)** |
| | *e2e* | 285(10.1%) | **26.4 (25.3%)** | **0.551 (22.5%)** | 1.01(10.9%) |
| | *S*: SchNet | 1237 | 62.2 | 0.214 | 0 |
| | *T*: PaiNN | 358 | 38.5 | 0.390 | 0.25 |
| *similar* | *Vanilla (1)* | **1139 (10.6%)** | **52.9 (13.1%)** | **0.2422 (16%)** | 0(0%) |
| | *Vanilla (2)* | 1140(10.5%) | 59.2(12.7%) | 0.241(15.1%) | 0(0%) |
| | *n2n* | 1170(7%) | 60(9.3%) | 0.235(11.9%) | 0(0%) |
| | *S*: PaiNN | 358 | 38.5 | 0.390 | 0.25 |
| | *T*: GemNet-OC | 226 | 22.5 | 0.61 | 1.89 |
| *different* | *Vanilla (1)* | 356(1.7%) | 38.3(1.1%) | 0.392(1.2%) | 0.258(0.5%) |
| | *Vanilla (2)* | 357(0.7%) | 43.5(-31.4%) | 0.334(-25.4%) | 0.210(-2.4%) |
| | *n2n* | **271 (66.0%)** | 37.3(7.5%) | 0.408(8.2%) | **0.477 (13.9%)** |
| | *e2n* | 330(21.8%) | **36.3(14.0%)** | **0.419 (13.4%)** | 0.371(7.4%) |
| | *v2v* | 369(-8.2%) | 37.2(8.0%) | 0.409(8.9%) | 0.217(-2.0%) |

Table 10: Evaluation results on the OC20 S2EF **out-of-distribution (adsorbates)** validation set.

| | Model | OC20 S2EF Validation (**out-of-distribution (adsorbates)**) | | | |
| | | Energy MAE meV ↓ | Force MAE meV/Å ↓ | Force cos ↑ | EFwT % ↑ |
|---|---|---|---|---|---|
| *same* | *S*: PaiNN-small | 469 | 47.9 | 0.334 | 0.03 |
| | *T*: PaiNN | 437 | 44.5 | 0.369 | 0.043 |
| | *Vanilla (1)* | 519(-156.2%) | 47.4(14.7%) | 0.334(-1.2%) | 0.003(0%) |
| | *Vanilla (2)* | 477(-23.6%) | 54.5(-106.2%) | 0.297(-109.9%) | 0.002(-76.9%) |
| | *n2n* | 443(80.9%) | **47.2 (19.8%)** | **0.337 (9.3%)** | 0.003(-10%) |
| | *v2v* | **441 (87.5%)** | 47.9(-1%) | 0.337(8.68%) | **0.04 (81.6%)** |
| | *S*: GemNet-OC-small | 325 | 31.0 | 0.521 | 0.190 |
| | *T*: GemNet-OC | 258 | 25.2 | 0.600 | 0.45 |
| | *Vanilla (1)* | 312(19.5%) | 31.0(-0.7%) | 0.522(2.1%) | 0.19(-0.9%) |
| | *Vanilla (2)* | 309(24.2%) | 30.9(1.4%) | 0.523(2.6%) | 0.20(2.7%) |
| | *n2n* | **282 (63.7%)** | 30.9(1.7%) | 0.523(5.8%) | 0.22(11.5%) |
| | *e2e* | 315(14.8%) | **29.3 (28.8%)** | **0.542 (26.3%)** | **0.23 (16.5%)** |
| *similar* | *S*: SchNet | 1344 | 58 | 0.196 | 0 |
| | *T*: PaiNN | 437 | 44.5 | 0.369 | 0.043 |
| | *Vanilla (1)* | 1247(10.7%) | 64.5(-49.7%) | **0.221 (14.7%)** | 0(0%) |
| | *Vanilla (2)* | **1245 (10.9%)** | 64.5(-48.2) | 0.22(14%) | 0(0%) |
| | *n2n* | 1286(6.3%) | 65(-51.9%) | 0.213(9.9%) | 0(0%) |
| *different* | *S*: PaiNN | 437 | 44.5 | 0.369 | 0.043 |
| | *T*: GemNet-OC | 258 | 25.2 | 0.6 | 0.45 |
| | *Vanilla (1)* | 424(7.2%) | 44.5(-0.2%) | 0.370(0.5%) | 0.052(2.3%) |
| | *Vanilla (2)* | 408(15.9%) | 49.2(-24.3%) | 0.315(-23.1%) | 0.036(-1.7%) |
| | *n2n* | **321 (64.9%)** | 43.2(6.9%) | 0.387(7.8%) | **0.084 (10.0%)** |
| | *e2n* | 407(16.9%) | **41.6 (15.3%)** | **0.498 (12.9%)** | 0.081(9.3%) |
| | *v2v* | 418(10.5%) | 42.0(13%) | 0.391(9.9%) | 0.058(3.7%) |

Table 11: Evaluation results on the OC20 S2EF **out-of-distribution (catalysts)** validation set.

| | Model | OC20 S2EF Validation (**out-of-distribution (catalysts)**) | | | |
| | | Energy MAE meV ↓ | Force MAE meV/Å ↓ | Force cos ↑ | EFwT % ↑ |
|---|---|---|---|---|---|
| *same* | *S*: PaiNN-small | 467 | 42 | 0.341 | 0.13 |
| | *T*: PaiNN | 412 | 39.2 | 0.369 | 0.23 |
| | *Vanilla (1)* | 466(4.9%) | 42.8(-28.4%) | 0.336(-16.9%) | 0.11(-20%) |
| | *Vanilla (2)* | 439(52.4%) | 45.4(-120.6%) | 0.306(-120.8%) | 0.12(-10%) |
| | *n2n* | **437 (56.3%)** | **42.0 (1%)** | **0.343 (8.2%)** | **0.14 (11.2%)** |
| | *v2v* | 444(43.4%) | 42.4(-12.8%) | 0.342(3.6%) | 0.12(-12%) |
| | *S*: GemNet-OC-small | 335 | 28.9 | 0.506 | 0.85 |
| | *T*: GemNet-OC | 288 | 24.0 | 0.576 | 1.68 |
| | *Vanilla (1)* | 339(-9.3%) | 29.0(-1.1%) | 0.507(0.9%) | 0.85(-0.4%) |
| | *Vanilla (2)* | 318(35.4%) | 28.9(-0.1%) | 0.507(1.0%) | **1.05 (24.1%)** |
| | *n2n* | **309 (54.4%)** | 28.8(2.0%) | 0.508(2.6%) | 1.02(20.5%) |
| | *e2e* | 324(21.6%) | **27.6 (26.5%)** | **0.524 (25.1%)** | 0.95(12.5%) |
| *similar* | *S*: SchNet | 1205 | 61.6 | 0.205 | 0 |
| | *T*: PaiNN | 412 | 39.2 | 0.369 | 0.23 |
| | *Vanilla (1)* | **1122 (10.6%)** | **58.7 (12.9%)** | **0.234 (15.9%)** | **0.01 (4.3%)** |
| | *Vanilla (2)* | 1122(10.5%) | 58.8(12.5%) | 0.23(15.2%) | 0(0%) |
| | *n2n* | 1150(6.9%) | 59.4(9.8%) | 0.225(12.3%) | 0(0%) |
| *different* | *S*: PaiNN | 412 | 39.2 | 0.369 | 0.23 |
| | *T*: GemNet-OC | 288 | 24 | 0.576 | 1.68 |
| | *Vanilla (1)* | 423(-8.8%) | 39.1(0.8%) | 0.371(1.1%) | 0.230(0.0%) |
| | *Vanilla (2)* | 400(9.5%) | 43.6(-29%) | 0.320(-23.5%) | 0.23(0.2%) |
| | *n2n* | **345 (54.0%)** | 38.5(4.7%) | 0.383(6.8%) | **0.433 (14%)** |
| | *e2n* | 401(8.9%) | **37.4 (11.9%)** | **0.395 (12.3%)** | 0.317(6.0%) |
| | *v2v* | 424(-10.7%) | 38.2(6.5%) | 0.386(8.2%) | 0.187(-3%) |

Table 12: Evaluation results on the OC20 S2EF **out-of-distribution (both)** validation set.

| | Model | Energy MAE meV ↓ | Force MAE meV/Å ↓ | Force cos ↑ | EFwT % ↑ |
|---|---|---|---|---|---|
| | | **OC20 S2EF Validation (out-of-distribution (both))** | | | |
| *same* | S: PaiNN-small | 610 | 56.8 | 0.346 | 0.02 |
| | T: PaiNN | 554 | 59.2 | 0.379 | 0.03 |
| | *Vanilla (1)* | 648(-67.6%) | 61.1(-179.2%) | 0.056(-900.3%) | 0.02(0%) |
| | *Vanilla (2)* | 592(32.2%) | 60.6(-158.3%) | 0.310(-112.4%) | 0.02(0%) |
| | *n2n* | 557(94.2%) | **56.0 (33%)** | 0.351(14.8%) | 0.018(-15.8%) |
| | *v2v* | **547 (112.6%)** | 56.5(12.3%) | **0.352 (17.3%)** | **0.025 (43.3)** |
| | S: GemNet-OC-small | 424 | 37.4 | 0.533 | 0.11 |
| | T: GemNet-OC | 370 | 31.0 | 0.606 | 0.23 |
| | *Vanilla (1)* | 412(23.1%) | 37.5(-1.8%) | 0.534(1.2%) | 0.11(-2.3%) |
| | *Vanilla (2)* | 401(43.2%) | 37.4(0.2%) | 0.535(2.0%) | 0.12(8.5%) |
| | *n2n* | **395 (53.7%)** | 37.3(1.6%) | 0.537(4.4%) | 0.12(8.5%) |
| | *e2e* | 412(22.2%) | **35.5 (29.5%)** | **0.554 (29.0%)** | **0.13 (19.9%)** |
| *similar* | S: SchNet | 1450 | 78.4 | 0.202 | 0 |
| | T: PaiNN | 554 | 59.2 | 0.379 | 0.03 |
| | *Vanilla (1)* | **1350 (11.2%)** | 75.9(13%) | **0.227 (14.3%)** | **0.0025 (1.8%)** |
| | *Vanilla (2)* | 1358(10.2%) | **75.7 (14.1%)** | 0.226(13.8%) | 0(0%) |
| | *n2n* | 1396(6.1%) | 76.2(11.5%) | 0.220(10.4%) | 0(0%) |
| *different* | S: PaiNN | 554 | 59.2 | 0.379 | 0.03 |
| | T: GemNet-OC | 370 | 31 | 0.606 | 0.23 |
| | *Vanilla (1)* | 558(-2.3%) | 53.8(19.0%) | 0.380(0.6%) | 0.031(-0.5%) |
| | *Vanilla (2)* | 511(23.3%) | 323.1(-935.7%) | 0.326(-23.2%) | 0.027(-2.6%) |
| | *n2n* | **448 (57.4%)** | 52.4(24.1%) | 0.394(2.4%) | 0.056(12.1%) |
| | *e2n* | 536(9.6%) | **50.1 (32.4%)** | **0.407 (12.5%)** | **0.057 (12.6%)** |
| | *v2v* | 538(8.5%) | 50.5(31.0%) | 0.402(10.4%) | 0.035(1.7%) |

# C  Baseline results on COLL

In Table 13, we present the performance and inference throughput of the baseline models on COLL. As the systems are much smaller than those in OC20, the throughput is a lot larger than the one observed in Table 2. Qualitatively, however, we observe the same, clear trade-off between accuracy and throughput.

Table 13: Evaluation of the performance of the four baseline models on the COLL dataset.

| Model | Inference Throughput Samples / GPU sec. ↑ | COLL test set Energy MAE meV ↓ | Force MAE meV/Å ↓ | Force cos ↑ | EFwT % ↑ |
|---|---|---|---|---|---|
| SchNet | 44000 | 146.5 | 121.2 | 0.970 | 2.75 |
| PaiNN-small | 29000 | 104.0 | 80.9 | 0.984 | 5.4 |
| PaiNN | 13000 | 85.8 | 64.1 | 0.988 | 10.1 |
| GemNet-OC | 3520 | 44.8 | 38.2 | 0.994 | 20.2 |

## D    Data augmentation

We investigated data augmentation as a way of distilling knowledge from GemNet-OC into PaiNN on the OC20 dataset.

### D.1    Data jittering

To create additional data, we added noise to the atomic positions of the training samples and then used the teacher to label the newly derived samples. We tried two different approaches: Random noise, and optimizing the positions using gradient ascent such that the difference between the predictions of the student and teacher was maximized as done in [45]. Denoting the noise as $\delta$, we obtain the noise atomic positions as $\boldsymbol{X}_\delta = \boldsymbol{X} + \delta$. Let the student and teacher models be denoted as $f_s$ and $f_t$ respectively, we then obtained the noise $\delta'$ by initializing this as $\boldsymbol{0}$ and

$$\mathcal{L}_{\text{KD}} = \mathcal{L}_0(f_s(\boldsymbol{X}_\delta, \boldsymbol{z}), f_t(\boldsymbol{X}_\delta, \boldsymbol{z}))$$
$$\delta' = \delta + \alpha \nabla_\delta \mathcal{L}_{\text{KD}}.$$

However, this becomes computationally expensive, as it requires additional gradients for $\nabla_\delta \mathcal{L}_{\text{KD}}$. Hence, we settled on using a single step, where we fixed the norm of $\delta$ to avoid going too far away from the real structure. We experimented with different norms, with the smallest being $0.1$ Å. We compared this to using random directions with a fixed norm, and we did not see any improvements when using the more computationally expensive gradient ascent approach. In both cases, the noise was added to all the samples in the batch.

### D.2    Synthetic data

**Combined dataset 2M+d1M.**    We generated 1M synthetic samples by first drawing 100k random adsorbate and catalyst combinations (systems) and then running relaxations with a pre-trained GemNet-OC model. Out of these relaxations with 100 steps on average (200 max), we randomly draw approx. 10% to obtain 1M samples.

In the next step, we combine the 1M samples with the 2M OCP dataset, which is based on DFT relaxations. Directly working with this combined dataset means iterating over a 1-to-2 ratio of samples from each subset in an epoch of 3M samples. To control this ratio, we define the target ratio of samples from the synthetic dataset during training $\alpha_{target} \in [0, 1]$. Setting $\alpha_{target} = 0.5$ means that per epoch we iterate over the 1M dataset 1.5 times and over the 2M dataset 0.75 times.

**Different weighting in loss depending on origin.**    Next to specifying a sampling ratio of samples from the synthetic dataset, we can also specify how to weight the contribution of samples to the loss based on their origin (DFT or synthetic). To achieve this, we specify the weighting ratio of synthetic to DFT samples $r_{\text{s/dft}} = w_{\text{s}}/w_{\text{dft}} \in \mathbb{R}^+$. In each batch, we compute a weighting factor in front of the synthetic $w_{\text{s}}$ and DFT $w_{\text{dft}}$ samples satisfying the conditions

$$w_{\text{s}} \cdot \alpha_{\text{batch}} + w_{\text{dft}} \cdot (1 - \alpha_{\text{batch}}) = 1, \tag{9}$$

where $\alpha_{\text{batch}}$ is the ratio of synthetic to DFT samples in a batch. Hence, when we combine DFT and synthetic data - $\alpha_{\text{batch}} \in (0, 1)$, we derive the following weights:

$$w_{\text{dft}} = (1 - \alpha_{\text{batch}} + \alpha_{\text{batch}} \cdot r_{\text{s/dft}})^{-1}, \tag{10}$$
$$w_{\text{s}} = r_{\text{s/dft}} \cdot w_{\text{dft}}. \tag{11}$$

Likewise, when $\alpha_{\text{batch}} = \{0, 1\}$ - i.e., we either train on DFT or synthetic data exclusively, the corresponding weighting coefficient ($w_{\text{s}}$ or $w_{\text{dft}}$) is naturally equal to $1$.

# E   Hyperparameter studies

We additionally investigated different aspects of our KD protocols, which we present below. We have performed these experiments when distilling GemNet-OC into PaiNN on the OC20-2M dataset.

## E.1   Effect of losses

In our experiments, we have used MSE as the loss $\mathcal{L}_{\text{feat}}$. However, in the general framework in Equation (3), there are other choices that are possible, e.g., more advanced losses like Local Structure Preservation [34] and Global Structure Preservation (GSP) [35]. We therefore initially experimented with using these alternative losses when distilling GemNet-OC into PaiNN on OC20. However, the initial experiments showed that MSE worked well, and in particular, a lot better than the more advanced GSP and LSP losses. We therefore settled on using MSE as our $\mathcal{L}_{\text{feat}}$. In Table 14, we present the average performance over all four validation splits when using these different losses in the *n2n* and *e2n* settings.

We also experimented with trying to optimize the CKA directly (as we saw that the CKA similarity improved when using distillation), but it did not work and we did not pursue it any further.

Table 14: Comparing different loss functions $\mathcal{L}_{\text{feat}}$ with GemNet-OC as teacher and PaiNN as student on OC20, using the *n2n* and *e2n* KD protocols. In the case of LSP + *e2n*, the model completely failed when predicting forces on the *ood (both)* validation set, leading to a force MAE of $464$ meV/Å, which is almost a factor 10 larger than the other models on the same split. We have therefore written this value as "-". When evaluating a checkpoint for a model which was trained half as long, the error on this split was $53.7$ meV/Å, and the average over all four validation splits was $44.8$ meV/Å.

|  | | OC20 validation set | | | |
|---|---|---|---|---|---|
| | Loss | Energy MAE meV $\downarrow$ | Force MAE meV/Å $\downarrow$ | Force cos $\uparrow$ | EFwT % $\uparrow$ |
| *n2n* | MSE | 346 | 42.8 | 0.393 | 0.262 |
| | GSP | 427 | 46.1 | 0.356 | 0.124 |
| | LSP | 398 | 44.8 | 0.367 | 0.159 |
| *e2n* | MSE | 430 | 41.3 | 0.405 | 0.195 |
| | GSP | 463 | 45.2 | 0.363 | 0.119 |
| | LSP | 441 | - | 0.380 | 0.134 |

## E.2   Effect of transformations

We have evaluated using different transformations $\mathcal{M}_{\text{s}}$, i.e., transformations of the student features before applying the loss $\mathcal{L}_{\text{feat}}$. We tried either using the identity function, (i.e., not using a transformation at all), a linear transformation (i.e., multiplication with matrix and adding a bias vector), or using a multilayer perceptron (MLP) with one hidden layer. We conducted our experiments when distilling G We found that using an MLP worsened the results, and for *e2n*, there was not a big difference between using a linear layer and no transformation at all. For *n2n*, the node features in PaiNN and GemNet-OC are of different dimensions, and we can therefore not use the identity transformation when using the MSE loss.

The results from the experiments are presented in Table 15.

## E.3   Effect of feature selection

GemNet-OC consists of an initial embedding layer, followed by a series of interaction layers. The result of each embedding/interaction layer is used as input into the next layer, while a copy is also processed by an "output layer". To make the final prediction, the results of the different output layers are concatenated and processed by a final MLP. This means that, for each embedding/interaction layer, we have two features that could potentially be distilled: the feature used as input for the next layer, or the result of the output layer. Additionally, we could use features from inside the final MLP which make the prediction by processing the concatenated output features.

Table 15: Comparing different transformation functions $\mathcal{M}$s (Identity, a linear layer and an MLP with one hidden layer) with GemNet-OC as teacher and PaiNN as student, using the *n2n* and *e2n* KD protocols. As the node features in GemNet-OC and PaiNN are of different dimensions, we cannot use the identity transformation when using *n2n*.

| | | OC20 validation set | | | |
|---|---|---|---|---|---|
| | Loss | Energy MAE meV ↓ | Force MAE meV/Å ↓ | Force cos ↑ | EFwT % ↑ |
| *n2n* | Identity | - | - | - | - |
| | Linear | 346 | 42.8 | 0.393 | 0.262 |
| | MLP | 363 | 45.1 | 0.367 | 0.147 |
| *e2n* | Identity | 430 | 41.3 | 0.405 | 0.195 |
| | Linear | 418 | 41.3 | 0.405 | 0.207 |
| | MLP | 427 | 43.2 | 0.387 | 0.161 |

Initially, we used the feature after the final interaction layer when distilling knowledge from GemNet-OC. However, we found that using the feature just before the final linear layer in the final MLP gave a drastic improvement in performance. We, therefore, set out to investigate how the choice of features impacted the results in more detail.

We performed these experiments when distilling GemNet-OC into PaiNN on OC20 using our *n2n* strategy, and the results presented here are on a set of 30 thousand samples sampled from the in-distribution validation set. We did not perform any extensive hyperparameter tuning, but chose $\lambda$ such that $\lambda \mathcal{L}_{\text{KD}}$ (the distillation loss term) was initially roughly the same for all choices.

**Choice of GemNet-OC layer.** In Figure 4, we present the training curves when fixing the choice of feature in PaiNN and varying the choice of features in GemNet-OC. The overall trend is that closer to the output is better: even using the features from the early output layers is better than using features from later interaction layers. Our results suggest that for forces, it is better to use features from earlier output layers. However, we think this could be due to the choice of $\lambda$, as we have empirically found that the weighting of the loss term in *n2n* could offer a trade-off between energy and force performance.

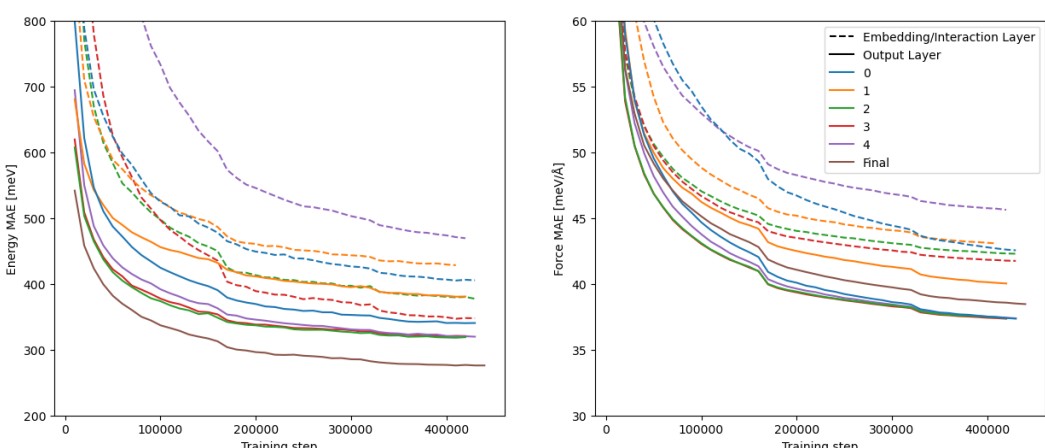

Figure 4: Evaluation error as we vary the features to distill from in the teacher model - energy MAE (*left*), and force MAE (*right*). *n2n* KD from GemNet-OC to PaiNN. Performance is evaluated on a validation subset comprising 30k samples. The numbers 0 to 4 indicate at what stage the feature has been extracted, with 0 meaning after the embedding layer, and 1 to 4 after the corresponding interaction layer. Solid and dashed lines indicate if the feature is the result of an embedding/interaction layer, or an output layer, respectively. "Final" refers to the feature extracted right before the final linear prediction layer.

**Choice of PaiNN layer.** PaiNN consists of a sequence of blocks, where each block consists of a message layer and an update layer. In Figure 5, we present training curves when fixing the choice of features in GemNet-OC (the feature just before the final linear layer), and varying the choice of features in PaiNN (choice of block, and either the feature after the corresponding message or update layer). The results here indicate that using deeper features leads to better results.

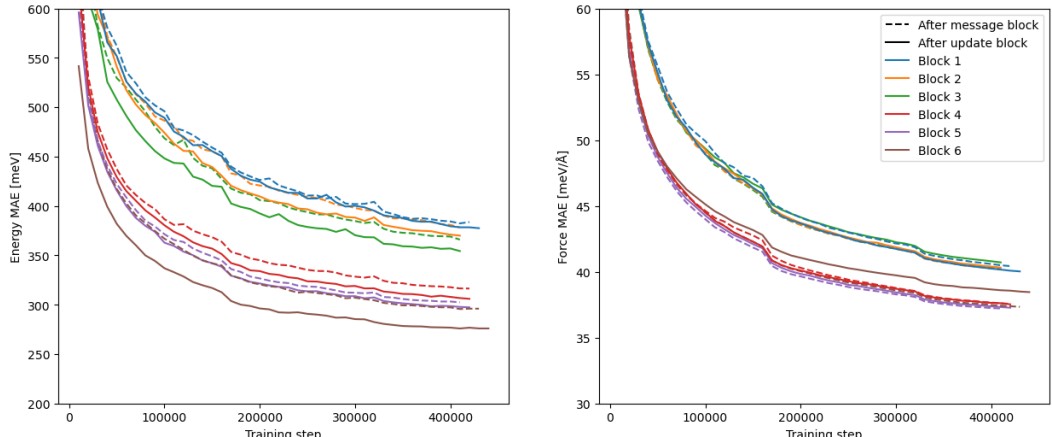

Figure 5: Evaluation error as we vary the features to distill into in the student model - energy MAE (*left*), and force MAE (*right*). *n2n* KD from GemNet-OC to PaiNN. Performance is evaluated on a validation subset comprising 30k samples. The different colors indicate after which block the features have been extracted, and dashed and solid lines indicate if features were extracted after the message or update layers, respectively.

**Conclusion.** The conclusion we draw from this study is that using features as close to the output as possible improves KD performance in our setup. However, these results are only empirical, and more investigation could be done. For example, if it is possible to beforehand determine which pairs of features should be used (and not having to rely on trial-and-error).

# F  Error bars

To get an idea of the stability of our KD protocols, we perform additional experiments distilling GemNet-OC into PaiNN using three different seeds and compute standard deviations. We present these results in Table 16.

Table 16: Performance of KD from GemNet-OC into PaiNN across 3 different seeds, averaged over all validation splits. The numbers are presented as mean $\pm$ one standard deviation. The missing force error for the baseline model is due to one of the seeds completely failing on the out-of-distribution (both) split, drastically increasing the error. The other two seeds had force MAEs of 43.8 and 45.3 meV/Å, respectively.

| | OC20 validation set | | | |
| --- | --- | --- | --- | --- |
| Loss | Energy MAE meV ↓ | Force MAE meV/Å ↓ | Force cos ↑ | EFwT % ↑ |
| None (baseline) | $440 \pm 8$ | - | $0.376 \pm 0.0018$ | $0.143 \pm 0.0051$ |
| n2n | $346 \pm 0.7$ | $43.2 \pm 0.6$ | $0.392 \pm 0.0017$ | $0.256 \pm 0.011$ |

# G Explainability

We utilize CKA similarity scores to monitor the effect of KD throughout our studies. Here, we present a selection of the analyses we have performed, summarizing how KD influences the similarity between teacher and student models across teacher-student configurations (Figure 6); across KD protocols (Figure 7); and feature selections (Figure 8). We found out that such similarity metrics can be effectively used to examine and profile different KD approaches, as well as as a potential debugging tool. We also explored the utility of CKA (in conjunction with measures of the predictive ability of individual features) as a means to inform the design of (optimal) KD strategies and feature selection protocols *a priori*, but the results were not conclusive to include here.

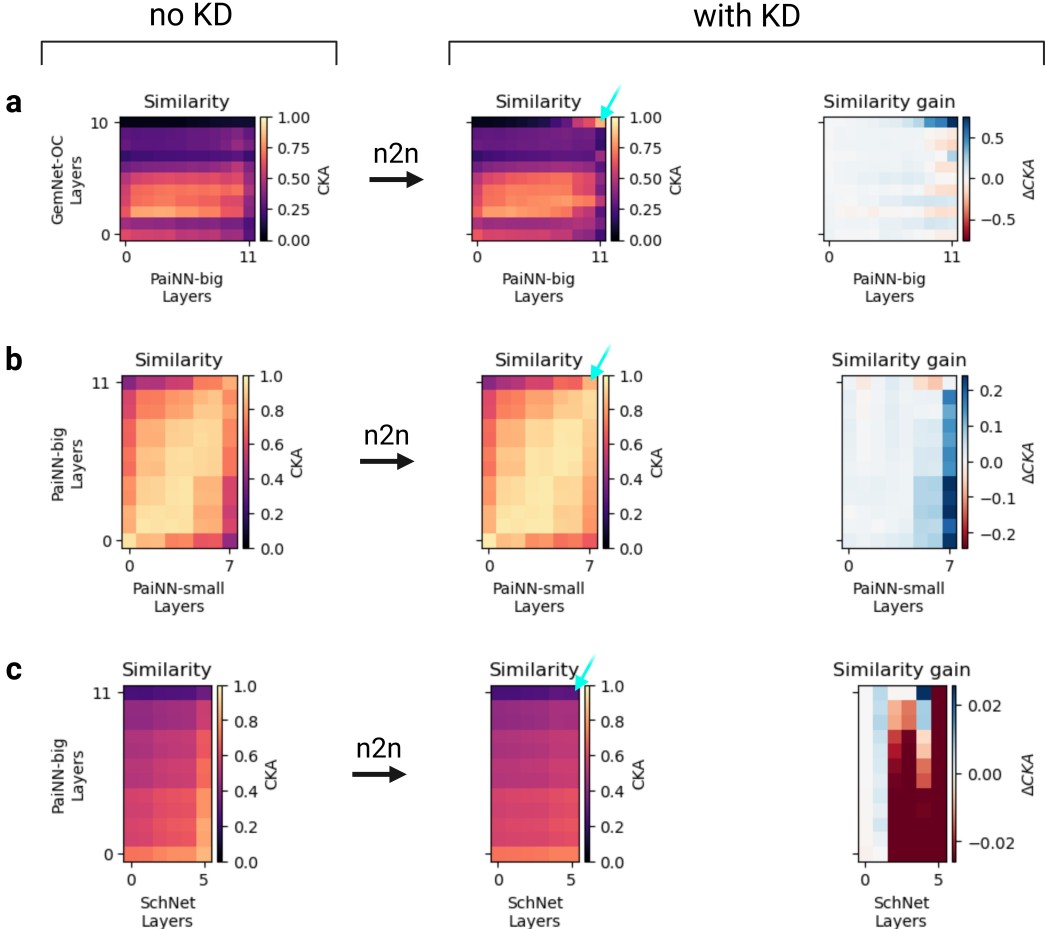

Figure 6: We explore the effect of *n2n* KD on the feature similarity between different student-teacher configurations: (a) GemNet-OC -> PaiNN; (b) PaiNN -> PaiNN-small; (c) PaiNN -> SchNet. The layer pair that was used in each experiment is indicated with a ⬈. Note the scale difference in the *Similarity gain* plots.

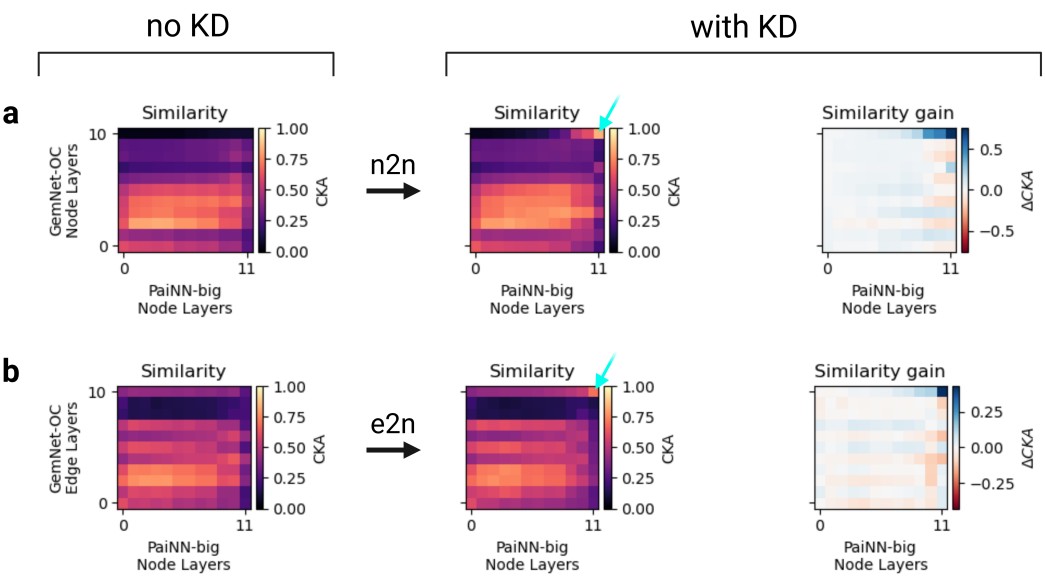

Figure 7: We explore the effect of different KD strategies - *n2n* and *e2n* KD on the feature similarity between the student and the teacher models. This is computed for GemNet-OC -> PaiNN: (a) *n2n*; (b) *e2n*. The layer pair that was used in each experiment is indicated with a ✓. Note the scale difference in the *Similarity gain* plots.

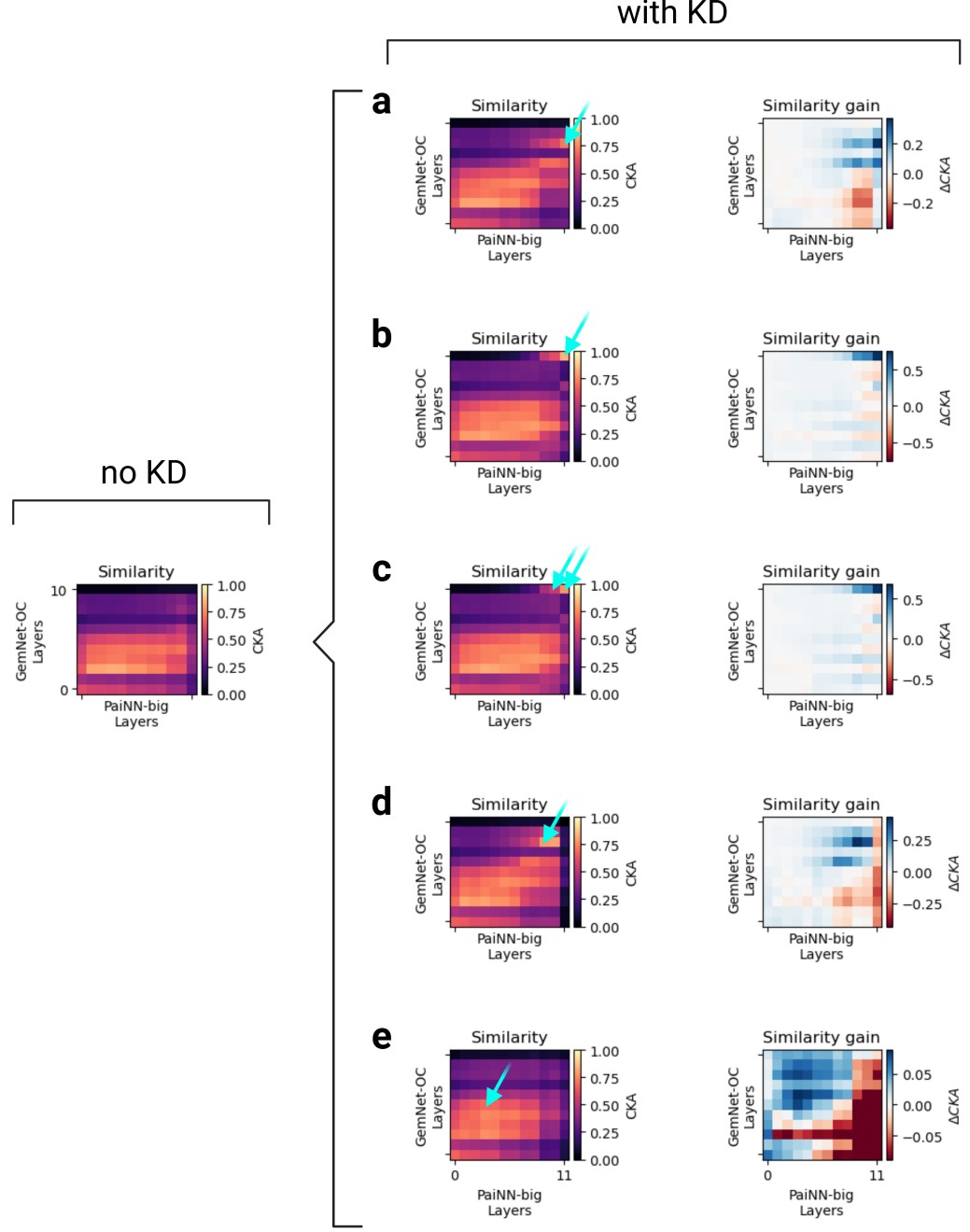

Figure 8: We explore the effect of feature selection in KD on feature similarity between the student and the teacher models: (a) H4->U6; (b) CONCAT+MLP->U6; (c) CONCAT+MLP->M6+U6; (d) H4->U5; (e) X2->M2. The layer pair that was used in each experiment is indicated with a ✐. Note the scale difference in the *Similarity gain* plots.

# H    Training times

One caveat of knowledge distillation is that it inherently increases the training time of the student model. In our offline KD setup, we need to perform additional forward passes through the teacher to extract representations to distill to the student. However, it is important to note that, despite increasing the computational time per training step, we observed that models trained with KD can outperform their baseline counterparts even when compared at the same training time point (Figure 9), despite the latter having been trained for more steps/epoch in total. This means that, all in all, we can use KD to enhance the predictive accuracy in models without necessarily impacting training times.

However, we make the following remark. In this experiment, we utilized publicly available pre-trained Gemnet-OC model weights, and therefore did not have to train the teacher model ourselves. However, when access to a pre-trained teacher model is not available, one should also account for the time required to train the teacher.

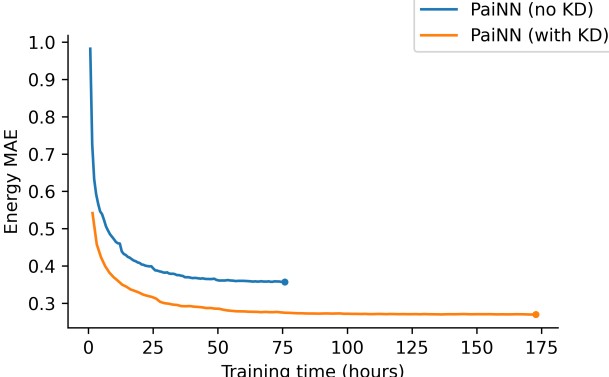

Figure 9: Energy validation error of PaiNN without (*blue*) and with (*orange*) knowledge distillation from GemNet-OC, trained for the same number of steps (1 million). Validation on a random sample of size 30k samples from the in-distribution OC20 validation set.

