# OpenReview forum: "Accelerating Molecular Graph Neural Networks via Knowledge Distillation"
_NeurIPS.cc/2023/Conference — NeurIPS 2023 poster_

### Official Review · Reviewer_6ZfK · 2023-07-02

**Soundness:** 3 good
**Presentation:** 3 good
**Contribution:** 3 good
**Rating:** 6
**Confidence:** 3

**Summary:**

* This paper explores knowledge distillation (KD) on speeding up molecular GNNs, which has challenges of being regression instead of classification, and having both scalar and vector outputs
* They tried a few different methods on a few different teacher-student combinations, with the best results closing 65% of the gap between the student and teacher in energy, and 21% in forces, slightly less for different student-teacher models.
* Analyzed performance with respect to similarity of models,
* Tried data augmentation techniques, but with no improvements

**Strengths:**

* This paper tackles an important problem that is an unstudied area: applying KD to GNNs of 3D space with regressions
* Tackles challenges of KD: regression instead of classification, vector outputs in addition to scalar
* Paper is clearly written, with good related background to my knowledge
* Despite the weaknesses described below, I believe this is an important area for KD to extend into, and this paper makes good initial progress at it.


**Weaknesses:**

* Despite good improvement for the small model for energy, it’s still quite a ways from the performance of the large model. Force metrics are especially bad, considering force MAE is almost 2x even with KD for some models.
* Beyond that, it is hard to get a sense of how good these improvements are. How does it compare to the improvement of KD in other fields? One suggestion is to plot speed vs. accuracy to see the tradeoff, which may help determine the usefulness in downstream applications (e.g. hypothetically, a small model is twice as fast but 10% less accurate, but perhaps you can make up for that by running a bit less than 2x the number of inference calls, so then that’s considered “good”).
* I would like to see other teacher-student combinations - for example, why 2 sizes of PaiNN but not other models? I am curious about big gemnet on small gemnet, since gemnet is the better model in this selection.


**Questions:**

* Why are forces much worse than energy improvement? Any analysis into this?
* Error bars? I’m not sure how much of Table 3 is noise
* I am unsure how much it matters about how similar the teacher and student model are. Isn’t it often that you use the same architecture for the student, just smaller?
* Why does vanilla KD do significantly worse on PaiNN-big to PaiNN-small? This doesn’t seem intuitive.
* Data augmentation: how much data did you add? Is the amount of rattling reasonable (e.g. too much movement may be OOD for the teacher)
* Suggestion for baseline: how good are your KD techniques when the student model is the exact same as the teacher?

**Limitations:**

The authors acknowledged limitations and potential negative impact.

---

> ### Author Rebuttal · Authors · 2023-08-09
>
> We want to thank the reviewer for their comments and feedback! We are extremely happy to hear they appreciate the importance of our work, the unique challenges we address in the paper, and the quality of the presentation. We also appreciate your insightful comments. We carefully respond to your concerns below.
>
> ### Performance improvements are substantial (W1)
> We respectfully disagree with the assertion that the improvements not being substantial. Using the KD strategies we study, ***we consistently close >60% of the gap*** between teacher and student models for energy predictions, and 10-20% for forces. Especially on the COLL dataset (which we now have moved to the main text) where these numbers go ***up to 96.7% and 62.5% for energy and force prediction, respectively*** (see the COLL table in the attached pdf).
>
> And still, we want to reiterate that this ***improvement is out-of-the-box*** with respect to inference throughput, which is the bottleneck we tackle in this paper. In other words, the improvement does not come at the expense of slower models at inference, meaning that even a small improvement could be useful. And moreover, ***this is the first work in the area***, and we hope we can inspire further work to further close the gap.
>
> #### Force predictions (W1+Q1)
> Given the amount of training data on energies (one scalar per system) vs forces (one vector per atom), we think that learning to predict energies is a harder problem. We hypothesize that by distilling knowledge from a teacher model "knowing" more about the energy surface of a molecule, ***there is more potential to improve the energy predictions***.
>
> Because of the tradeoff between energy and force accuracy, people often develop 2 separate models optimized for energy and force predictions, tuning different hyperparameters (Gasteiger *et al.* (2021)). Here, we do not perform any such optimization specifically. But, we still achieve force improvements up to 62.5%, which combined with our energy prediction gains, constitute a good overall improvement for our initial exploration in the area. Future work can focus on better optimizing for force predictions.
>
> ### Putting performance improvements into context (W2)
> We believe this is hard to quantify as previous research is mostly on classification, where percentage gains are normally smaller. To put the performance into some context, closing almost 60% of the gap in energy predictions between GemNet-OC and PaiNN on OC20 gives an energy error lower than that of GemNet-dT (which is severely slower), trained on the same OC20-2M dataset (Gasteiger *et al.* (2022)).
>
> We thank the reviewer for the suggestion to plot speed vs. accuracy. We created such a plot, but due to the different scales, the performance was less visible and we instead plotted in the format of Figure 1, which conveys the same information. We also have that information in Table 2, which summarises the tradeoff between the models. We now edited Figure 1 to include the results on the COLL dataset and benchmarked the models on COLL, which we added to the appendix.
>
> ### Importance of model similarity (Q3)
> We can see two aspects in the question:
> 1) Our results do not in the end show that model similarity is important for KD.
> 2) You were already under the impression that model similarity shouldn't be a necessity for successful KD before our analysis.
>
> Regarding 1), we think that this is a contribution of our work; we initially thought that it would be easier to distill between more similar models, but the results suggest otherwise.
>
> About 2), we were not of this impression before seeing the final results, and although model similarity doesn't seem to be a requirement for knowledge distillation, our results (e.g. Figure 3) indeed indicate that model similarity increases with KD.
>
> ### Vanilla KD on PaiNN-big to PaiNN-small (Q4)
> Note that Vanilla KD appears to not do well only on OC20 between PaiNN-big to PaiNN-small. We believe that is because the gap between the force prediction accuracy of the 2 baseline models used in this scenario is already rather small, meaning the extra signal we provide during Vanilla KD is not as significant.
>
> ### Data augmentation (Q5)
> In the case of random rattling, we added noise to all samples in the batch. To avoid adding too much noise (and going OOD), we experimented with adding noise of a fixed norm, and we tried different values starting from 0.1 Å. We have added this information in the appendix.
>
> For the synthetic data, we mixed it with the real data, with the fraction of synthetic data in a single batch being \{0%, 5%, 10%, 20%, 50%, 80%\} on average.
>
> ### Error bars (Q2)
> We thank the reviewer for the comment. We have run additional runs for GemNet-OC -> PaiNN on the OC20 dataset. A table can be found in the attached pdf, also added to the appendix and we add a reference in the results section. The force results for the baseline model are missing, as one of the runs completely failed on the ood_both task. The force MAEs for the baseline were 45.3, 43.7, and 115.7 meV/Å.
>
> ### Other teacher-student combinations (W3)
> We agree that distilling into a smaller GemNet-OC model would be an interesting experiment. However, given the limited time, we have not been able to finalize these experiments. Preliminary, it looks like energy predictions see a substantial improvement when using the n2n loss.
>
> ### Baseline suggestion (Q6)
> We thank the reviewer for the suggestion for another baseline, but we are not sure what the reviewer means here by *"exact same"* - distilling to the exact same model configuration - e.g. PaiNN-big to PaiNN-big. That would be considered teacher-free KD which we haven't explored, but seems to be an established and well-performing method [1].
>
> [1]Yuan et al, Revisiting Knowledge Distillation via Label Smoothing Regularization, CVPR 2020

---

> > ### Comment · Reviewer_6ZfK · 2023-08-18
> >
> > Thank you for the thorough response.
> >
> > > We respectfully disagree with the assertion that the improvements not being substantial.
> >
> > To clarify, it is great that you close 60% of the gap, but it's not the improvement of the gap I'm concerned about, it's the absolute performance of the student model. While this paper makes great progress on improving the small models, it may be that the small models are still unusable due to such a high force/energy MAE - for example, the best student force MAE in Table 3 is 42.1meV, much worse than large GemNet-OC at 25.7. It's easy to have an impressive % of closing the gap when the gap is large to begin with.
> >
> > > we are not sure what the reviewer means here by "exact same"
> >
> > Yes, I meant the exact same model configuration. The purpose of this would be to study the effect of the distillation process during training, independently of the difference in model size/architecture, and it would be good to see similar results as in [1]. (This is a minor suggestion that is more of a sanity check that the KD process is working as expected)

---

> > > ### Author Response · Authors · 2023-08-19
> > >
> > > Thank you for your response and for engaging in the discussion! We respond to any remaining concerns below.
> > > ___
> > > >While this paper makes great progress on improving the small models, it may be that the small models are still unusable due to such a high force/energy MAE
> > >
> > > The accuracy of a model is not the only consideration for downstream applications. Applications typically rather care about the trade-off between speed and accuracy. If they would not, they could just run the full DFT calculation. This is demonstrated well by model families in other, more developed fields such as vision (e.g. the EfficientNet model family) or language (e.g. the Llama model family). **Smaller molecular GNN models can be useful** for e.g. pre-screening materials, simulations on "easy" in-distribution data, or for running in tandem with a larger model that provides corrections when needed.
> > >
> > > As such, KD is a method for **pushing the Pareto frontier in the speed vs. accuracy space**. The student models are indeed less accurate, but they are also **3x and 8x faster**.
> > >
> > > For KD, it is most interesting to explore configurations where the student is substantially faster, which typically comes with a similar downside in accuracy. This implies a challenging problem: **Closing a large percentage of a large gap means that the absolute improvement is also large**.
> > >
> > > >It's easy to have an impressive % of closing the gap when the gap is large to begin with.
> > >
> > > We understand your perspective, but think that the opposite might also be true: obtaining a large relative improvement of a large gap can be more challenging than achieving the same when the initial gap is small, since the former is associated with a larger absolute improvement. **Either way, we provide examples for both cases, as we experiment with teacher and student models that have variable gaps in performance.**

---

### Official Review · Reviewer_n7yW · 2023-07-04

**Soundness:** 3 good
**Presentation:** 3 good
**Contribution:** 2 fair
**Rating:** 7
**Confidence:** 5

**Summary:**

This paper aims to improve the performance of resource-efficient GNNs for molecular simulation, an area where top models are becoming increasingly larger and more cumbersome.

Several knowledge distillation approaches are proposed with the aim of regressing the smaller student model's embeddings onto those of a larger teacher model.

The paper conducts a unified empirical benchmark of various distillation approaches and demonstrates that smaller molecular GNN performance can be non-trivially boosted when trained using KD from a larger teacher model.


**Strengths:**

## Significant motivation to accelerate GNNs for molecular simulation
- Improving scalability and efficiency of GNNs specialized for molecular simulations is a worthwhile research question. The best architectures for this area are increasingly becoming very large in terms of compute requirements, so improving the performance of smaller models is certainly worthwhile.
- I agree that this is the first paper to propose distillation as an approach to boost smaller models - the novelty of the overall contribution also makes this paper interesting.

## Experiments are performed in a fair and unified manner under the same pipeline
- This is true to the best of my understanding and without having looked at the code.
- Empirically benchmarking and demonstrating to what extend various KD techniques can boost molecular GNNs is worthwhile.

## Overall clear and well-structured presentation

**Weaknesses:**

## Limited technical novelty
- Equation 3 essentially projects teacher and student node features to a common dimension and performs regression to align the embedding spaces. And perhaps this is 'all we need' for doing effective KD for molecular GNNs.
- However, in my opinion, this technical idea is of limited novelty.
    - I will caveat this by saying that empirical studies which may not introduce new methodology are very important. I would not reject a paper for this. I think the empirical benchmark is a strength.
- I don't think anything about Equation 3 is very specific to GNNs for molecular simulation. While this paper may be the first to apply and evaluate these ideas for molecular simulations (which is worthwhile), I don't think the technical ideas are sufficiently novel or tailored to this area.

## Missing contextualization w.r.t. general GNN distillation literature
- The study is certainly valuable for its focus on molecular simulation applications. However, better contextualization and (possibly) comparison with existing work on general purpose GNN representation knowledge distillation could further improve it.
- The introduction (line 37) claims that KD is limited for regression tasks, but feature-based KD or representation distillation is a generally applicable technique regardless of what downstream task one is interested in. I currently do not see a good reason to not compare to at least some standard baseline on general purpose representation distillation, or (better) to existing work on GNN representation distillation ([LSP, Yang et al.](https://openaccess.thecvf.com/content_CVPR_2020/papers/Yang_Distilling_Knowledge_From_Graph_Convolutional_Networks_CVPR_2020_paper.pdf), [G-CRD, Joshi et al.](https://arxiv.org/abs/2111.04964)). Joshi et al. could also be cited for introducing CKA-based embedding similarity analysis in the context of GNNs, which this paper's analysis also uses.
- Line 87 states that distillation approaches for standard GNN architectures have only concentrated on classification tasks. I do not agree with this claim, since representation distillation approaches are agnostic to the downstream task.

**Overall, I felt that the paper does not give enough credit to existing, more general purpose solutions on GNN distillation in the literature. Yes, the goal of the paper is to be an empirical study specific to molecular simulations and this is a positive (see Strengths). However, the proposed techniques are seemingly not bespoke to molecular simulation, so better contextualization and comparison to existing work is warranted, in my opinion.**

**Questions:**

- The Introduction or elsewhere in the Experiments could provide a key quantification of whether/how much the proposed techniques boosted GNNs over training from scratch and baseline KD techniques. Figure 1 gives a nice illustration, but it is unclear which KD technique was used to obtain each of the three results. In each of the cases, was it the baseline or one of yours?

- For line 136, please elaborate a bit on what 'complications' we need to account for? Why can't we just use the scalar component of features to perform distillation in these networks? (As my reading of this paper's results seem to suggest that distilling from vector channels was not actually that useful...)

- For Table 2's EFwT column, an application-oriented question: Yes, we can distill into small and efficient models like SchNet and boost their performance slightly, but is this scientifically relevant? For instance, would you use the SchNet model after your distillation based training for running molecular simulations?

- The formulation of v2v via equations 3 and 6 was a bit worrying to me. If the student's final vector feature is regressed onto the vector feature of the teacher using mean absolute/square error, won't doing this not account for how these quantities are equivariant/geometric vectors in 3D space?
    - For example, there could be a very high MSE between two rotated versions of the same set of vectors. However, if you found the optimal alignment and then computed MSE, it would be zero.
    - Thus, comparing two sets of geometric vectors requires first finding an optimal alignment/rotation matrix and then computing the RMSE. Just computing unaligned RMSE may not be very informative. This is standard practice in fields such as protein structure prediction.
    - Have I misunderstood this aspect of the work?

**Limitations:**

There is some discussion on limitations and potential negative societal impact. I would have been interested in deeper discussions around both the technical aspects as well as application-specific limits of this line of work.

For instance, whether GNN model expressivity, especially for geometric and equivariant GNNs, plays a role in the choice of teacher-student pairings? There may be classes of functions that certain student models can provably not learn ever (eg. distinguishing types of neighborhoods or entire geometric graphs). The paper does not propose any principles for selecting teacher-student pairings for molecular simulations. (This is related to my question on whether a distilled SchNet is ever really useful in practice, too.)

Additionally, for these OCP datasets, could it be interesting to distill across tasks? What may happen if we distill from S2EF models into IS2RE models (a task with significantly lesser data), and is this interesting? Did you try this?

---

> ### Author Rebuttal · Authors · 2023-08-09
>
> We are grateful for the detailed feedback! Happy to hear you appreciate the importance and novelty of our work, and the quality of our empirical results.  We also appreciate your insightful comments. We carefully respond to your concerns below.
>
> ### Technical novelty of our work and contextualization (W1+W2)
> We agree that putting our work into context is central and made multiple improvements to address this gap.
>
> We have ***improved the contextualization*** in our paper based on the discussion points below (see general response for the exact changes).
>
> Following the suggestion of the reviewer, we ran ***additional experiments with LSP and GSP (see table in attached pdf)***.
>
> We added a reference to Joshi et al. (2021) as a paper also using CKA.
>
> We furthermore want to highlight that:
> 1) Eq. 3 represents a general framework for feature-based KD, which we agree is not new on its own. KD research is focused on finding new methods based on that framework-i.e finding appropriate representations (neurons, layers, combinations of layers, relationships); transformations (projections, fusion, fission, etc), and losses (MSE, MAE, LSP, GSP, G-CRD). Thus, not every new method that belongs to the class of feature-based KD is not novel. Actually, GSP and G-CRD methods (Joshi et al. (2021)) mentioned are new types of KD loss, which also fall under the framework defined by Eq 3.
> 2) Here, we do not create a new KD loss but tackle the question of what type of features to distill in molecular GNNs (a separate, unique degree of freedom in our context). We define and study 5 strategies for distilling different features across molecular GNNs. They are indeed built upon the overarching framework of feature-based KD, but adapted to the requirements of this field, e.g. how to respect the physics of the problem, which types of features to use, how to transform equivariant and directional representations.
> 3) As such, they are not directly comparable to methods like LSP, and G-CRD. Actually, our strategies can be regarded as general frameworks that can use LSP, GSP, G-CRD as a KD loss. In our study, we decided to use MSE loss as we found it more appropriate for our setup. Contrastive learning (the backbone of LSP, GSP and G-CRD) is useful for node classification where you want to contrast different classes of nodes, but less clear how to use for graph-level regression tasks on molecules (i.e. atoms are a part of the whole). We had already experimented with GSP as a loss, available in the appendix where we perform ablation studies of layers, transformations and losses. MSE seemed to work significantly better.
> 4) We agree that as a general approach, feature-based KD is mostly agnostic to the downstream task. Yet, ***prior work on applying feature-based KD to regression task is quite limited. Most feature distillation methods are tailored towards classification tasks*** - e.g. still use logit information (e.g. LSP, GSP, and G-CRD).
>
> ### Definition of v2v KD (Q4)
> We believe this is a misunderstanding. Note that the vectorial features of *both* the student and the teacher are equivariant by construction, and PaiNN uses this to make equivariant force predictions. It is hence desirable that also the directions of the vectorial features are correct, and rotating the vectors before applying the loss would not encourage this. It is therefore ***important to use a loss that encourages vectors to align in both magnitude and direction***. To make that more clear in the text, ***we add the following in connection to Eq. 6***: "Note that as these features are equivariant to rotations, it is important to use a loss that encourages vectors to align in both magnitude and direction - e.g. MSE."
>
> ### Experimental results are available in Table 3 (Q1)
> We are puzzled by the comment that we should: "provide a key quantification of whether/how much the proposed techniques boosted GNNs over training from scratch and baseline KD techniques.". We provide detailed experimental results in Table 3 (and Table 12 for COLL), which give all the details mentioned in this comment. Could you clarify what you think is missing?
>
> We indeed summarize these in Fig 1, where we depict the gap between student and teacher models. We are happy the reviewer found this figure useful. We agree that the caption of Figure 1 was not clear enough. Now it explicitly says that percentages represent our best results for each teacher-student configuration depicted.
>
> ### Clarifying line 136 (Q2)
> We apologize for any confusion here. ***We have made that more explicit:*** by adding *"as one needs to extract and align representations corresponding to comparable features in both models."*
>
> ### SchNet is scientifically relevant (Q3)
> We see the point with respect to OC20. But it is likely relevant for cases with a well-sampled chemical space. Also, simple, well-established methods protect us from overfitting to highly-engineered models. We decided to use SchNet as it is still one of the most widely known models in the field.
>
> ### Discussing limitations
> We thank the reviewer for the very helpful suggestions as potential limitations.
> - ***"defining teacher-student pairings based on their (provable) expressivity"*** - understanding the theory behind KD is a very active area of research (how it works, when it works, etc.). It is definitely an interesting future research avenue, so ***we added that to future work***.
> - ***"distillation across OCP tasks"*** - This is not something we have considered, but we think you have a point that this extra data could be used. However, we don't envision this within a **KD framework**, but rather a **pretraining framework**. The reason for this is that the two tasks are very different, and we think it is unclear how the knowledge of a teacher should be distilled into a student.
>
> We have ***expanded our discussion into limitations/future work*** and added the aforementioned points, as well as other suggestions from the reviewers.

---

> > ### Comment · Reviewer_n7yW · 2023-08-16
> > **Questions clarified; score increased**
> >
> > Thank you for the rebuttal. My questions have mostly been clarified and I'm happy to increase my score to reflect this.
> >
> > ---
> >
> > > Most feature distillation methods are tailored towards classification tasks - e.g. still use logit information...
> >
> > I still disagree. By definition, no feature based distillation methods use the logits. They use the features to perform distillation.
> >
> > > Definition of v2v KD (Q4): We believe this is a misunderstanding. Note that the vectorial features of both the student and the teacher are equivariant by construction...
> >
> > Thank you for the clarification, understood.
> >
> > > Could you clarify what you think is missing?
> >
> > Perhaps which KD method was the one used for each of the improvements in Figure 1. But overall, I agree. The interested reader can just spend some time reading through the results table to figure the same thing out.

---

> > > ### Author Response · Authors · 2023-08-19
> > >
> > > Thank you for your response and for engaging in the discussion! We are very happy to see that our improvements and arguments have addressed your concerns and made a substantial difference in your evaluation!
> > >
> > > ___
> > > >By definition, no feature based distillation methods use the logits. They use the features to perform distillation.
> > >
> > > Yes, we fully agree that, as a general method, the distillation of features is not targeted to a specific downstream task. However, we note that most methods proposed in the literature have been developed and/or evaluated with classification in mind, including the aforementioned LSP, GSP, and G-CRD, which also make use of logit information through their additional vanilla KD term.

---

### Official Review · Reviewer_xfxb · 2023-07-08

**Soundness:** 3 good
**Presentation:** 3 good
**Contribution:** 2 fair
**Rating:** 4
**Confidence:** 3

**Summary:**

This paper investigates the use of knowledge distillation (KD) to accelerate molecular dynamics using graph neural networks (GNNs) and improve their predictive accuracy. They describe their experiments with various KD protocols such as node to node, edge to node, and vector to vector KD for four different molecular GNN architectures, namely, PaiNN, SchNet, GemNet-OC. They demonstrate that KD can improve the speed with minor compromise on accuracy of molecular GNNs without altering their architecture by studying the OC20 catalysts.

**Strengths:**

The paper demonstrates the approach of knowledge distillation for developing faster GNN-based interatomic potentials for molecular simulations. This is an important problem to be addressed in the field.

**Weaknesses:**

There are several weaknesses for the paper as outlined below.

1. The authors have selected three different GNNs as the teacher models. However, the performance of the GNNs are not comparable to the state-of-the-art for the same architectures on the same dataset (if I am not mistaken) as confirmed by other publications and OC20 dashboard.
2. Authors selected only dataset for the evaluation, which is OC20. However, OC20 is not primarily a molecular simulation dataset as I understand. It is a dataset for energy and force prediction primarily based on DFT minimization. Although the task of force and energy prediction is similar molecular simulations, the dynamics task is not tested. Other more complex datasets on dynamics are not included in testing.
3. Although, this is one of the first attempts for KD in molecular simulations, the improvement in performance of most of the models is only marginal and not substantial.
4. The stability of these models on molecular simulation is not tested. Further, error evaluation metrics are not exhaustively presented [1].

[1] Fu, X., Wu, Z., Wang, W., Xie, T., Keten, S., Gomez-Bombarelli, R. and Jaakkola, T., 2022. Forces are not enough: Benchmark and critical evaluation for machine learning force fields with molecular simulations. arXiv preprint arXiv:2210.07237.

**Questions:**

1. The choice of the specific architectures is not clear. Although the authors have tried to be extensive by the selection of three different architectures, none of these architectures are giving SOTA performance. Accordingly, the reasons for choosing these architectures are not clear. It would be interesting to try the approaches on SOTA architectures such as equiformer, NequIP, or allegro.
2. The baselines chosen seem to be not really fair and give mostly negative results which suggests that they are incapable of any knowledge distillation. Fair baselines, should be implemented based on regression tasks and appropriate loss functions.
3. More extensive studies on different and challenging datasets should be carried out. This is available in most of the papers on machine learned potentials such as 3BPA, AcAc, alanine dipeptide, or LiPS.
4. Stability of the KD-based model need to be analyzed more carefully to understand the limitations and applicability of the models.

**Limitations:**

Authors have not included a detailed discussion on the limitations of the study. This should be included.

---

> ### Author Rebuttal · Authors · 2023-08-09
>
> We thank the reviewer for their feedback and are glad to hear that they acknowledge the importance of the problem we are tackling! We also appreciate your insightful comments. We carefully respond to your concerns below.
>
> ### We have an additional benchmark dataset: COLL (W2+Q3)
> We wish to highlight that ***we have actually run experiments on 2 distinct datasets, including a molecular dynamics dataset - COLL!*** ***Please see the discussion in our general response.***
>
> ### OC20 is a highly relevant molecular simulation dataset (W2+Q3)
> We are confused about the statement that OC20 is not a molecular simulations dataset - ***it is a dataset concerned with the simulation of catalyst-adsorbate systems***. In this work, we tackle the task of predicting energy and forces, which is the most general task in OC20 and has the broadest applicability across catalysis and related fields (Chanussot *et al.* (2021)). Also, note that performing DFT minimization in high throughput searches is of high interest in the materials science domain, and inference speed is of high importance to enable covering as large a part of chemical space as possible. ***OC20 is the largest dataset for this task***, and we thus think this is a highly relevant dataset.
>
> ### The models we study were SOTA at the time of research (W1+Q1)
> We also want to highlight that ***the models investigated in this paper were SOTA on both OC20 and COLL at the time of research and experimentation***. Looking at the relevant S2EF leaderboard, even today there are only three models better than GemNet-OC: SCN, eSCN and EquiformerV2. However, note that the public weights of SCN and eSCN were only released in February and March this year, in parallel to our work (Aug 2022-May 2023), and the weights for EquiformerV2 were released in July 2023 (after our submission). Hence, GemNet-OC was SOTA at the time of our work (***and even today is still very representative of the SOTA***). The other models we study (PaiNN and SchNet) are chosen as they represent models that are faster, but not SOTA performance. ***To make that more clear to the reader***, we include that information in the text.
>
> ### Performance improvements are not marginal (W3)
> We respectfully disagree with the assertion that most of the improvements we achieve are marginal. Using the KD strategies we study, ***we consistently close >60% of the gap*** between teacher and student models for energy predictions, and 10-20% for forces. Especially on the COLL dataset, which the reviewer might have missed (we hope this is now more apparent after moving these results to the main text), where these numbers go ***up to 96.7% and 62.5% for energy and force prediction, respectively*** (refer to the COLL table in the one-page pdf document attached to our response).
>
> And still, we want to reiterate that this ***improvement is out-of-the-box*** with respect to inference throughput, which is the bottleneck we tackle in this paper. In other words, the improvement does not come at the expense of slower models at inference, meaning that even a small improvement is still useful. Moreover, ***this is the first work in the area***, and we hope we can inspire more work to further close the gap.
>
> ### Vanilla KD are fair and not arbitrary (Q2)
> We are not sure why our 2 vanilla KD approaches have come across as unfair to the reviewer.  Our vanilla KD methods are ***not arbitrary***. These are strategies inspired by vanilla KD from classification, adapted to our context and undergone significant optimization as part of our work.We agree that the results we achieve with vanilla KD are not always perfect, but ***they do prove promising in many experiments, often outperforming other techniques***. ***Especially on the COLL dataset***, which the reviewer might have missed (we apologize again and hope this is more apparent now with the results moved to the main text), where looking at force prediction, Vanilla (2) outperforms the other methods on 2 of the 3 teacher-student configurations, and achieves just slightly worse than n2n on the third.
>
> Also, we want to highlight that there might have been a misunderstanding about the inclusion of the two vanilla KD approaches in our study due to how we had originally phrased some parts of the text. The two vanilla KD strategies ***have not been investigated in this field before***, so analyzing their performance is as much part of our work as are the other strategies. In other words, these are additional strategies we study, and not really baselines we want to compare with or beat necessarily.
>
> However, we agree that this may not be clear from the phrases we use in the paragraph where we define these two methods. ***We make that more clear in the text (see general response)***.
>
> ### Discussing limitations and future work
> We agree that our previous discussion on limitations/future work was somewhat narrow, only concerning potential issues around increasing training times. Accordingly, ***we have expanded this section and provided a more comprehensive account of the limitations of our methods and future work***, mentioning future directions like combining KD strategies (e.g. n2n and v2v); extending the framework to other types of features, molecular tasks and datasets; improving the theoretic intuition behind KD; and performing a more comprehensive stability analysis of the approach.
>
> ### Stability analysis and error evaluation metrics (W4+Q4)
> We concur that a stability analysis would be indeed valuable. However, considering that this is a first work on KD for molecular GNNs, we have used well-established metrics, and have categorized stability analysis as beyond the present research. We thank you for the suggestion and have ***included this aspect in our limitations and future work discussions***.
>
> In terms of error evaluation metrics, we do provide all the metrics that are relevant for the datasets and tasks we benchmark on - OC20 and COLL.

---

> > ### Comment · Reviewer_xfxb · 2023-08-18
> > **Thank you for the response**
> >
> > > We have an additional benchmark dataset: COLL (W2+Q3)
> >
> > Thank you for pointing the attention to the COLL dataset and adding them to the main manuscript.
> >
> > > OC20 is a highly relevant molecular simulation dataset (W2+Q3)
> >
> > I am not questioning the relevance of the dataset. The main issue was that the experiments were performed only on one dataset, which focused on one task. There are several papers on MD potentials etc., and there is a reason why they demonstrate it on different datasets instead of one. Because the tasks associated with each dataset are different, it is important to show whether the approach works in diverse situations. For instance, whether a trained potential is generalizable to unseen temperatures (3BPA), compositions (rMD17), pressures, and sizes (in terms of the number of atoms, for example, LiPS or water), and whether it can capture bond-breaking (AcAc dataset), etc. It is not clear whether the student potential has these capabilities.
> >
> > > Performance improvements are not marginal (W3)
> >
> > I still disagree with this. I think showing percentage improvement is not necessarily a good way to argue in this case, as another reviewer pointed out, when there is a huge percentage difference is there, to begin with. I think the question would be whether they are acceptable potentials or not, and for this, stability analysis is important. As of now, it is not clear whether the new potential is truly usable or not.
> >
> > > The models we studied were SOTA at the time of research (W1+Q1)
> >
> > Again, I disagree. NequIP, allegro, MACE, BotNet, etc. were all released in 2022 and much before the submission. Equiformer V1.0 was released in ICLR 2022. It is possible that they did not test it on the OC20 dataset, a specific model that the authors have chosen. But these models are demonstrated on several other datasets. This also highlights why choosing one particular dataset and relying on the experiments on this dataset is not necessarily a good approach

---

> > > ### Author Response · Authors · 2023-08-19
> > >
> > > Thank you for your response and for engaging in the discussion! We are happy to hear that your concerns about us having 1 dataset have been addressed after highlighting our results on the additional COLL dataset. We respond to your remaining concerns below.
> > >
> > > ### **Generalization and diversity**
> > > We agree that it is important to show whether the approach works and is generalizable in diverse situations. This is actually why we have selected to do our analyses on the OC20 and COLL datasets, in particular. **OC20 is by far the largest and most chemically diverse benchmark.** It thoroughly tests generalization to unseen system sizes and compositions, with 3/4 of the validation set being out-of-distribution data. **COLL is also a diverse benchmark dataset**, comprising highly distorted structures at high temperature, including bond formations. As such, we do believe we demonstrate extensive validation and generalization, sufficient to support our goal to showcase the potential of KD in the area.
> > >
> > > ### **Evaluation metrics and stability analysis**
> > > We also fully agree that long-term stability is an interesting and important research direction. However, it is also important to note that stability analyses are quite involved and computationally expensive. This is why **stability analysis** has been investigated mainly in cases where it is crucial, e.g. when looking at statistical properties over long simulation rollouts, and **is not something that is routinely done in papers similar to ours**.
> > >
> > > **Almost all of the published work in the area relies on the same metrics we use** and does not perform the suggested stability analysis, including very recent papers [1,2,3]. In this work, **we used common, established error metrics**, which are the standard for datasets like OC20, COLL, and others. As in other works, we think this is sufficient to demonstrate that our methodology is promising. Still, to recognize the significance of stability analyses, we include this as future work in our paper.
> > >
> > > [1] Passaro et al., Reducing SO(3) Convolutions to SO(2) for Efficient Equivariant GNNs, ICML 2023
> > >
> > > [2] Duval et al., FAENet: Frame Averaging Equivariant GNN for Materials Modeling, ICML 2023
> > >
> > > [3] Lioa et al., EquiformerV2: Improved Equivariant Transformer for Scaling to Higher-Degree Representations, 2023
> > >
> > > ### **Performance improvements**
> > > The accuracy of a model is not the only consideration for downstream applications. Applications typically rather care about the trade-off between speed and accuracy. If they would not, they could just run the full DFT calculation. This is demonstrated well by model families in other, more developed fields such as vision (e.g. the EfficientNet model family) or language (e.g. the Llama model family). **Smaller molecular GNN models can be useful** for e.g. pre-screening materials, simulations on "easy" in-distribution data, or for running in tandem with a larger model that provides corrections when needed.
> > >
> > > As such, KD is a method for **pushing the Pareto frontier in the speed vs. accuracy space**. The student models are indeed less accurate, but they are also **3x and 8x faster**.
> > >
> > > For KD, it is most interesting to explore configurations where the student is substantially faster, which typically comes with a similar downside in accuracy. This implies a challenging problem: **Closing a large percentage of a large gap means that the absolute improvement is also large**.
> > >
> > > We understand your perspective that percentage improvements can potentially be easier to achieve when there is an already huge percentage difference, but we think that the opposite might also be true: obtaining a large relative improvement of a large gap can be more challenging than achieving the same when the initial gap is small, since the former is associated with a larger absolute improvement. **Either way, we provide examples for both cases, as we experiment with teacher and student models that have variable gaps in performance.**
> > >
> > > And we highlight percentages of closing the gap because this is _exactly_ what KD tries to do - i.e. to reduce the gap between models.
> > >
> > > ### **SOTA models**
> > > We agree that there are many other state-of-the-art models that would be interesting to investigate. However, this work is not about benchmarking a specific model or proposing a new SOTA, but about showing that KD is a promising general method in this area.
> > >
> > > As such, **we evaluate KD on well-performing, established architectures**: GemNet-OC was the best _available_ model on the OC20 leaderboard, and PaiNN and SchNet are widely-used, more lightweight models. Importantly, these models cover a wide diversity of approaches, and KD still works consistently: GemNet-OC is based on edge representation and angles, PaiNN on vectorial representations, and SchNet on node representations.
> > >
> > > Additional models would of course make the evidence even stronger, but we think that the presented consistent improvements already show the potential of KD in this area.

---

### Official Review · Reviewer_t1R8 · 2023-07-11

**Soundness:** 3 good
**Presentation:** 3 good
**Contribution:** 3 good
**Rating:** 7
**Confidence:** 2

**Summary:**

The main contribution of this work is the exploration of knowledge distillation as a means to enhance the performance and scalability of GNNs for molecular simulations. The authors introduce custom KD strategies, namely node-to-node, edge-to-node, and vector-to-vector
 distillation, to overcome limitations of KD for regression tasks in GNNs. The performance of the KD protocols is evaluated by training student models to predict molecular properties such as energy and forces. The results show improvement in the performance of student models.


**Strengths:**

The paper is well written and the topic is of practical importance.

**Weaknesses:**

See questions.

**Questions:**

It would have been interesting seen these methods applied to model with Tensor features like e3nn, which are much heavier to train than the models in the paper.

**Limitations:**

The authors discuss the limitation of their approach.

---

> ### Author Rebuttal · Authors · 2023-08-09
>
> We thank the reviewer for the concise, positive review! We are glad to hear that they appreciate the presentation of our work and acknowledge its importance!
>
> ### Applying KD to tensorial models
> We also thank the reviewer for their suggestion to apply our methods to tensorial models like e3nn, but we kindly regard that as beyond the scope of this paper. The focus of this work was to conduct an empirical evaluation of the utility of KD in the area, which we show across 3 different architectures that represented the SOTA at the time of experimentation. Recognizing the positive findings of our work, we agree that it would be an interesting extension to also explore how they translate to tensorial models. To carry this message to the reader, ***we have added this as an additional point in our discussion on future work***.

---

### Official Review · Reviewer_o8B6 · 2023-07-15

**Soundness:** 2 fair
**Presentation:** 3 good
**Contribution:** 2 fair
**Rating:** 5
**Confidence:** 3

**Summary:**

the paper proposes new knowledge distillation strategies to enhance the hidden representation in the molecular GNN, downstream regression task of energy and force predicition shows promising results

**Strengths:**

1. the paper is well written and easy to follow
2. the experiments show the knowledge distillation can improve the regression performance.

**Weaknesses:**

1. to handle various structures in the molecule dataset, the paper proposes to use some GNN model to extract hidden features and do feature-based KD on top of that. the novelty is limited.
2. need to include more benchmark datasets to demonstrate the effectiveness. Also, in table 3, vanilla KD can outperform the proposed method in certain scenarios.

**Questions:**

see above

---

> ### Author Rebuttal · Authors · 2023-08-09
>
> We thank the reviewer for their feedback. We are glad to hear that you like the presentation of our work and highlight the improvement in performance we achieve! We also appreciate your insightful comments. We carefully respond to your concerns below.
>
> ### We have an additional benchmark dataset: COLL (W2.1)
> We would like to highlight that we have actually run experiments on ***2 distinct datasets*** - ***OC20-2M*** (the biggest and most diverse dataset in the area) and ***COLL*** (a challenging dataset for molecular dynamics). Due to space constraints, we had to put the results on COLL in the appendix, which might have made them easy to miss. We now moved these results into the main paper. ***Please see the discussion in our general response.***
>
> ### Vanilla KD is part of our investigation, not a baseline to beat (W2.2)
> We believe there might have been a misunderstanding about the inclusion of the two vanilla KD approaches in our study due to how we had originally phrased some parts of the text. The two vanilla KD strategies (inspired by the vanilla, logit-based KD used in classification problems (Hinton *et al.* (2015)) ***have not been investigated in this field before***, so analyzing their performance is as much part of our work as are the other strategies. In other words, these are additional strategies we study, and not really baselines we want to compare with and beat necessarily. As such, the fact that they can sometimes outperform n2n, e2n, and v2v does not undermine our contribution - to show that KD is a viable strategy in the context of molecular GNNs and compare different methods.
>
> However, we agree that this may not be clear from the phrases we use in the paragraph where we define these two methods. ***To make that more clear in the text, we change***:
> - the caption of the paragraph from *"Baseline KD strategies"* to *"Additional KD strategies"*;
> - and the introduction of that paragraph from *"To validate the performance of our KD strategies, we evaluate their performance against 2 vanilla-based KD approaches suitable for regression tasks."* to *"We further evaluate two additional KD approaches inspired by the vanilla logit-based KD used in classification, which we augment to make suitable for regression tasks."*
>
> ### Novelty of our work (W1)
> The reviewer does not mention a reason behind their assertion that our work is of limited novelty, so we are not sure what the problem may be. ***We believe our work is a novel contribution to the field because:***
> - *We introduce KD to molecular modelling*: we explore the utility of KD for molecular GNNs for the first time;
> - *Technical novelty*: we adapt feature-based KD to explore custom KD strategies (n2n, e2n, v2v, 2xVanilla) for the distillation of representations in equivariant and directional molecular GNNs. These are indeed built upon the established framework of feature-based KD, but they have been significantly adapted to the requirements of this field, e.g. how to respect the physics of the problem, which features to use, how to perform distillation).
> - *Comprehensive empirical analysis*: we conduct extensive empirical analyses and ablation studies of our framework and its components (features to distill, transformation functions, losses, data augmentation methods).
>
> As such, it perfectly ***aligns with the definition of novelty outlined in NeurIPS' guidelines***.

---

> > ### Comment · Reviewer_o8B6 · 2023-08-19
> > **Thanks for the response**
> >
> > I read the whole rebuttal and appreciate authors' response. I increased my rating.

---

### Author Rebuttal · Authors · 2023-08-09

We wish to take this opportunity to thank the reviewers for their time and effort in assessing our work. We are deeply grateful for the reviews we have received. We were extremely happy to hear that the reviewers recognized the importance of our work and the quality of our results and presentation!

We also appreciate the reviewers' insightful comments and feedback, which significantly contributed to the improvement of our presentation. We have taken the time to carefully respond to all of their concerns separately. Below, we include some of the central points of the reviews, which we believe are important to clarify.

### Additional benchmark dataset: COLL
We would like to highlight that we have actually run experiments on ***2 distinct datasets*** - ***OC20-2M*** (the biggest and most diverse dataset in the area) and ***COLL*** (a challenging dataset for molecular dynamics). We apologize if our ***results on COLL***, originally presented in Table 12 in the appendix due to space constraints, escaped the reviewer's attention. Recognizing the importance of such additional validation, ***we have decided to move Table 12 to the main text*** and extend our discussion of the results on the COLL dataset. We have ***also revisited Figure 1 to include results on COLL*** to make the presence of an additional benchmark dataset more apparent. We have also included the table summarizing our results on COLL in the one-page pdf document attached to our response. Note that we had previously had an error in the percentage calculations for GemNet-OC -> PaiNN-big on COLL, resulting in significantly smaller percentage improvements than those actually achieved. We have fixed that and updated the table.

### Clarification on Vanilla KD strategies
We believe there might have been a misunderstanding about the inclusion of the two vanilla KD approaches in our study due to how we had originally phrased some parts of the text. The two vanilla KD strategies (inspired by the vanilla, logit-based KD used in classification problems (Hinton *et al.* (2015)) ***have not been investigated in this field before***, so analyzing their performance is as much part of our work as are the other strategies. In other words, these are additional strategies we study, and not really baselines we want to compare with and beat necessarily. As such, the fact that they can sometimes outperform n2n, e2n, and v2v does not undermine our general line of work - to demonstrate that KD is a viable strategy in the context of molecular GNNs.

However, we agree that this may not be clear from the phrases we use in the paragraph where we define these two methods. ***To make that more clear in the text, we change***:
- the caption of the paragraph from *"Baseline KD strategies"* to *"Additional KD strategies"*;
- and the introduction of that paragraph from *"To validate the performance of our KD strategies, we evaluate their performance against 2 vanilla-based KD approaches suitable for regression tasks."* to *"We further evaluate two additional KD approaches inspired by the vanilla logit-based KD used in classification, which we augment to make suitable for regression tasks.*

### Other changes to the paper:
- ***Additional experiments (see one-page pdf)***
    - Additional seeds for GemNet-OC -> PainNN-big (see response to reviewer 6ZfK)
    - Additional losses, LSP and GSP (see response to reviewer n7yW)
    - Additional results GemNet-OC -> GemNet-OC-small (currently training - see response to reviewer 6ZfK)
- ***We have revisited parts of our methods section to improve the contextualization*** of the proposed techniques (see response to reviewer n7yW). More notably, we have added the following to the definition of *n2n*: *"Note this is a general approach that utilizes node features only, making it applicable to standard GNNs. Here, we want to enforce the student to mimic the representations of the teacher for each node (i.e. atom) independently, so we use a loss that directly penalizes the distance between the features in the two models, such as MSE (similar to the original formulation of feature-based KD in Romero et al. (2014)). Other recently proposed losses L_feat for the distillation of node features in standard GNNs specifically include approaches based on contrastive learning (Yang et al. (2021), Joshi et al. (2021), Yu et al. (2022) Huo et al. (2022)) and adversarial training (He et al. (2022)). We do not focus on such methods as much since they are better suited for (node) classification tasks (e.g. contrasting different classes of nodes), and not for molecule-level predictions."*
- ***We have expanded our discussion on limitations and future work*** to accommodate the suggestions of the reviewers, mentioning future directions like combining KD strategies (e.g. n2n and v2v); extending the framework to other types of features, molecular tasks, and datasets; improving the theoretic intuition behind KD and its applicability; and performing a more comprehensive stability analysis of the approach."

---

### Decision · Program_Chairs · 2023-09-21

**Decision:**

Accept (poster)

**Comment:**

Throughout the reviews, most of the reviewers agree that this submission contributed a dedicated empirical study of a range of methods on the distillation for molecular force field models which provides evidence for future research, while there are also concerns on e.g., technical novelty and statement on related work, absolute error and usability of the distilled model, and significance on the choice of the models and evaluation datasets/metrics. In follow-up discussions with the reviewers, they finally tend to accept this work considering that the empirical study on distillation methods on molecular tasks could be a worthy contribution. I hence recommend an accept. Nevertheless, the authors should be aware of and try to address the concerns that the limited number of datasets and not using SOTA models (e.g., GemNet (https://arxiv.org/abs/2106.08903), NequIP (https://www.nature.com/articles/s41467-022-29939-5)) weaken the solidity of the empirical study, and that the intention of distillation is not clearly described. For the latter point, the authors should respond to "if the intention is a lighter-weighted model to run molecular dynamics, modeling chemical reactions, or finding stable structures, then the method should be evaluated on these workloads, for which the sacrificed accuracy may or may not corrupt the stability". Ultimately, the method is for an application in the science domain but not a purely machine learning task. The authors should respect that in the paper.